# AlphaGRPO: Unlocking Self-Reflective Multimodal Generation in Unified Multimodal Models via Decompositional Verifiable Reward

**Runhui Huang** [1]  **Jie Wu** [2]  **Rui Yang** [1]  **Zhe Liu** [1]  **Hengshuang Zhao** [1]

## Abstract

In this paper, we propose **AlphaGRPO,** a novel framework that applies Group Relative Policy Optimization (GRPO) to AR-Diffusion Unified Multimodal Models (UMMs) to enhance multimodal generation capabilities without an additional cold-start stage. Our approach unlocks the model's intrinsic potential to perform advanced reasoning tasks: Reasoning Text-to-Image Generation, where the model actively infers implicit user intents, and Self-Reflective Refinement, where it autonomously diagnoses and corrects misalignments in generated outputs. To address the challenge of providing stable supervision for real-world multimodal generation, we introduce the Decompositional Verifiable Reward (**DVReward**). Unlike holistic scalar rewards, DVReward utilizes an LLM to decompose complex user requests into atomic, verifiable semantic and quality rubrics, which are then evaluated by a general MLLM to provide reliable and interpretable feedback. Extensive experiments demonstrate that AlphaGRPO yields robust improvements across multimodal generation benchmarks, including GenEval, TIIF-Bench, DPG-Bench and WISE, while also achieving significant gains in editing tasks on GEdit without training on editing tasks. These results validate that our self-reflective reinforcement approach effectively leverages inherent understanding to guide high-fidelity generation. Project page: `https://huangrh99.github.io/AlphaGRPO/`

## 1. Introduction

Recent advancements in Unified Multimodal Models (UMMs) focus on designing unified architectures capable of seamlessly integrating visual understanding and genera-

tion (Team, 2024; Wang et al., 2024; Xie et al., 2024; Wang et al., 2025a; Niu et al., 2025a; Huang et al., 2025a; Deng et al., 2025; Xie et al., 2025), marking a distinct shift from pure AR to hybrid AR-Diffusion architectures. Distinct from specialized models, these unified models possess the innate capability to process interleaved multimodal inputs and outputs. Crucially, this structural unification endows them with the potential to orchestrate complex cognitive workflows within a single end-to-end model, encompassing reasoning, execution, self-reflection, and refinement. However, effectively reinforcing UMMs to leverage their intrinsic understanding to improve multimodal generation remains a largely unexplored challenge.

Reinforcement Learning (RL), notably Group Relative Policy Optimization (GRPO) (Shao et al., 2024), has demonstrated remarkable success in reinforcing reasoning capabilities in LLMs (Shao et al., 2024; Guo et al., 2025) and optimizing visual generation in flow-matching diffusion models (Liu et al., 2025a; Xue et al., 2025). To enable complex tasks like reasoning text-to-image generation or self-reflective refinement, recent works (Deng et al., 2025; Wu et al., 2025b; Huang et al., 2025b) primarily rely on proprietary models to synthesize high-quality data. Although effective, this paradigm inevitably introduces an additional cold-start SFT stage, implying that the performance gains might stem from the distillation of the stronger teacher models. In contrast, we argue that since unified models already acquire fundamental primitives and implicit reasoning-related data through large-scale pretraining, it is possible to activate and enhance these dormant capabilities using RL without the cold-start stage.

The success of applying GRPO in multimodal generation relies on a reward model yielding stable, robust signals. To enhance broad, real-world multimodal generation capabilities, such a reward model is required to accurately assess diverse real-world samples. However, current visual generation RL often overlooks this, chasing high scores on training-aligned metrics (Liu et al., 2025a; Xue et al., 2025). This risks reward overfitting and fails to guarantee consistent improvements across diverse downstream benchmarks. In the pursuit of a universal evaluator, Multimodal Large Language Models (MLLMs) have emerged as the premier

---

[1]The University of Hong Kong [2]Bytedance Seed. Correspondence to: Hengshuang Zhao <hszhao@cs.hku.hk>.

*Proceedings of the 43rd International Conference on Machine Learning*, Seoul, South Korea. PMLR 306, 2026. Copyright 2026 by the author(s).

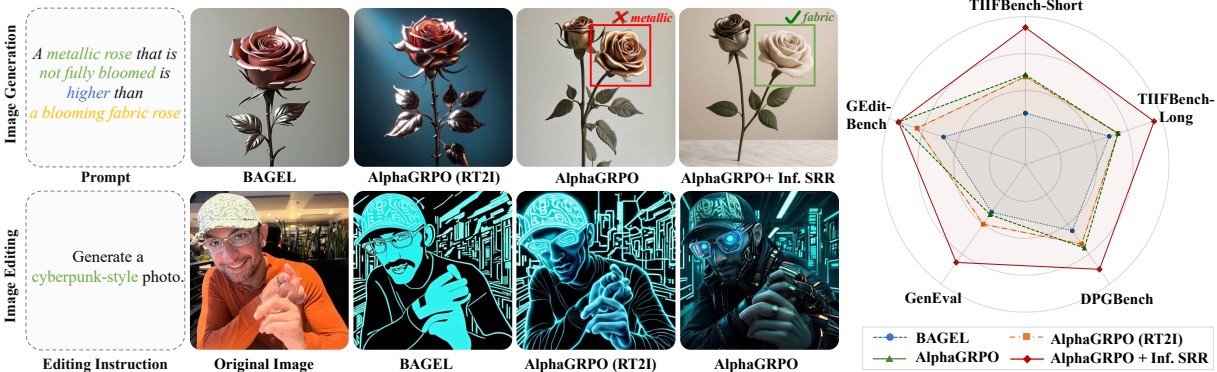

*Figure 1.* **Qualitative and quantitative comparisons of AlphaGRPO.** In Text-to-Image (top), our AlphaGRPO(trained on self-reflective refinement (SRR)) exhibits superior initial composition compared to Bagel, while applying Inference-time Self-Reflective Refinement (Inf. SRR) further rectifies fine-grained attribute mismatches (e.g., the "metallic" correct to "fabric" textures). In Image Editing (bottom), while the BAGEL baseline fails to capture the "cyberpunk" style, without training on the image editing task, AlphaGRPO trained on reasoning text-to-image (RT2I) generation demonstrates effective improvement on this task, and AlphaGRPO trained on SRR masters the style transformation. Quantitative comparison demonstrates that AlphaGRPO could consistently improve across five downstream benchmarks.

candidates, due to their robust understanding capabilities and extensive world knowledge. Fine-tuning these models on human preference datasets can yield specialized reward models with improved alignment accuracy (Wang et al., 2025b; Ma et al., 2025; Xiong et al., 2025). However, it shifts the model's distribution towards a limited domain and implicitly narrows the MLLM's capacity to handle open-world samples. Therefore, it becomes crucial to explore the stable, high-quality reward signals from general MLLMs without compromising their inherent understanding.

In this paper, we propose AlphaGRPO, a novel framework that extends GRPO to multimodal generation in AR-Diffusion UMM. It enhances unified multimodal understanding and generation capabilities by unlocking the model's intrinsic potential, without an additional cold-start stage. Specifically, we formulate multimodal generation as the unified trajectory that first generates text, then the image. We focus on the self-reflective refinement, which requires autonomously diagnosing misalignments from the initial generation results and executing correction strategies. This process demands a comprehensive synergy of capabilities, including multimodal perception, understanding, and generation. We introduce the False-Positive Rectification to eliminate the false improvement signals during training. Furthermore, we apply AlphaGRPO to reasoning text-to-image generation to validate the generalizability and robustness of AlphaGRPO across diverse multimodal tasks. To ensure reliable reward signals and promote robustness in real-world scenarios, we introduce the Decompositional Verifiable Reward (DVReward). This mechanism utilizes an LLM to decompose complex user requests into atomic, verifiable rubrics and verify them against the generated visual content using MLLM confidence scores.

In our experiments, we prioritize evaluating the method's generalization ability across diverse downstream tasks,

rather than relying on in-distribution test sets. As illustrated in Figure 1, powered by DVReward, AlphaGRPO training on both reasoning T2I (RT2I) and self-reflective refinement consistently improves performance on image generation and image editing benchmarks. Furthermore, in the Self-Reflective Refinement task, without training on editing data, AlphaGRPO not only maintains comparable gains to Alpha-GRPO (RT2I) on image generation benchmarks and secures a 0.52 improvement on editing benchmark, i.e., GEdit (Liu et al., 2025b), validating generalizability. Moreover, leveraging the inference-time self-reflective refinement further elevates T2I performance, reaching 83.9% on TIIF-Bench and outperforming Bagel by 5.8%.

The contributions of this paper can be summarized:

- We propose AlphaGRPO, the first framework to introduce GRPO training to AR-Diffusion Unified Models. By eliciting the model's latent primitives without an additional cold-start stage, we enable advanced capabilities in both Reasoning Text-to-Image Generation and Self-Reflective Refinement.

- We introduce Decompositional Verifiable Reward (DVReward), a novel fine-grained reward mechanism that decomposes user prompts into atomic verifiable rubrics across both semantic alignment and visual fidelity. This approach provides stable, interpretable supervision signals for multimodal generation GRPO training that indicate the correct way to use MLLM as the reward model.

- Our experiments demonstrate that Alpha-GRPO achieves consistent and significant improvements across multimodal generation benchmarks (e.g., GenEval, TIIF-Bench) and multimodal editing tasks (e.g., GEdit), proving the effectiveness and generalizability of AlphaGRPO.

## 2. Pilot Study

Before detailing our methodology, we conducted a pilot study to investigate two fundamental premises essential for aligning unified Multimodal Large Language Models (MLLMs): (1) whether pretrained UMMs possess the latent reasoning patterns required for self-reflective refinement and how to activate this, and (2) whether current MLLMs can provide reliable, discriminative reward signals to evaluate visual generation in open-world scenarios.

**Explicit Error-Seeking Activates Latent Reasoning.** To explore how to activate latent reasoning, we probe the state-of-the-art UMM, Bagel, with two tasks: Verification, where the model judges whether misalignments exist between the generated image and the user prompt, and Reflection, where the model is told that the image contains mistakes and is asked to diagnose them. Our experiments reveal a critical failure in verification: as shown in Figure 2, the model struggles to correctly identify obvious errors, instead frequently asserting that the image effectively fulfills the user's original intent. This indicates a pervasive confirmation bias (Huang et al., 2023), where the model easily assumes the generated content is correct. Conversely, when switched to Reflect Mode, it effectively breaks this confirmation loop and the model successfully scrutinizes details to identify issues with the shadow's position. This empirical finding demonstrates that the reflection mechanism maximizes the activation of UMMs' intrinsic visual understanding, providing a critical supervision signal to assist generative tasks. Building on this insight, we leverage this mechanism as the core foundation of our proposed Alpha-GRPO, specifically designing the framework to reinforce Self-Reflective Refinement capabilities during training.

**Asking Questions Yields Discriminative Reward Signals.** A reliable reward model should give a discriminative score for the images with nuanced differences against the input prompt. To assess the reliability of MLLMs as reward models, we generated two images based on the same prompt "A tree in front partially hides a bench behind it", where the first image fails to meet the spatial requirement while the other succeeds, as illustrated in Figure 3. We then compared two scoring mechanisms using Qwen3-VL-30B-A3B (Bai et al., 2025). First, we employed a Holistic Scalar Reward, VIEScore (Ku et al., 2024), directly prompting the model to assign a quality score (0-10) to the images and normalize the score to 0–1. The results reveal a critical limitation: the model assigns an identical score of 0.848 to both the failed and successful images, indicating that the model struggles to provide discriminative values when asked for an abstract assessment. To further investigate the capability of MLLM to distinguish the images, we directly ask the question about the key spatial attribute from the prompt (e.g., "Does the tree partially hide the bench?") and require the model to

*User:* A hand is holding a pair of scissors. The hand and scissors are casting a shadow onto the wall behind them. The scene is a sunny outdoors image of a hand, scissors, and shadow on a wall.

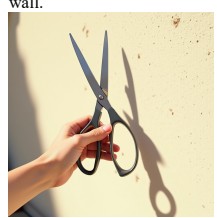

**Verification** — Fail to find error!

*Response*: The generated image **effectively fulfills the user's original intent.** It accurately depicts a hand holding scissors, casting a shadow on a wall in a sunny outdoor setting, with high visual quality and adherence to physical rules.

**Reflection** — Find error!

*Response*: **The image has issues with the shadow's position and shape**, which are inconsistent with the light source and the expected shadow pattern of the scissors and hand. To improve the image, **the shadow should be adjusted to be consistent with the light source and the position of the scissors**. The shadow on the wall should also be corrected to have a more natural and coherent appearance.

*Figure 2.* **Comparison of verification and reflection behaviors in UMMs.** We instruct BAGEL to verify (judging whether mistakes appear) or reflect (tasked with finding the mistakes) on the generated image. The "Reflect" mode activates the UMM's understanding ability to correctly identify the error.

*Prompt:* A tree in **front** partially hides a bench behind it.

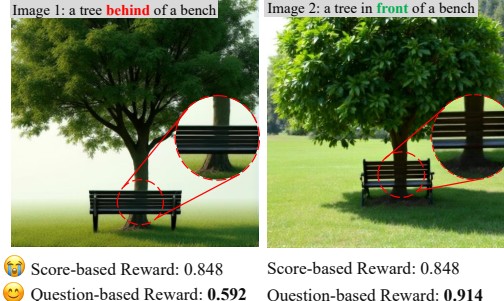

Image 1: a tree **behind** of a bench    Image 2: a tree **front** of a bench

Score-based Reward: 0.848      Score-based Reward: 0.848
Question-based Reward: **0.592**    Question-based Reward: **0.914**

*Figure 3.* **Comparison of Score-based vs. Question/Rubric-based Rewards.** Given two images generated from the prompt *"A tree in front partially hides a bench behind it"*, Image 1 fails the spatial constraint while Image 2 succeeds. The **Question-based Reward** (querying *"Does the tree partially hide the bench?"* via 'Yes' token logits) yields discriminative scores that correctly reflect the quality difference. In contrast, the **Score-based Reward** (VIEScore (Ku et al., 2024)) assigns identical scores to both, failing to distinguish the semantic error.

answer Yes or No. Instead of asking for a score, we calculate the probability of the "Yes" token . This method yields a highly discriminative signal (0.592 vs. 0.914), accurately reflecting the superior alignment of the second image. These findings imply that while holistic scalar scoring acts as a "black box" that smooths over semantic discrepancies, probing the model with specific questions via token logits effectively activates its discriminative capabilities. This finding motivates the design of our Decompositional Verifiable Reward, which provides the stable reward signals necessary for effective GRPO training.

## 3. Preliminary

In this section, we review the Group Relative Policy Optimization (GRPO) algorithm (Shao et al., 2024) and its distinct formulations for discrete language modeling and continuous visual generation tasks.

**GRPO for Language Modeling.** GRPO (Shao et al., 2024) was initially introduced for Large Language Models (LLMs) in mathematical reasoning tasks to eliminate the critic model required by PPO (Schulman et al., 2017), instead estimating the baseline from group scores. Given a query $q$, we sample a group of $G$ text outputs $\{\mathbf{y}_i\}_{i=1}^G$ from the behavior policy $\pi_{\theta_{old}}$. The optimization objective is:

$$\mathcal{J}_{\text{GRPO}} = \mathbb{E}\left[\frac{1}{G}\sum_{i=1}^{G}\frac{1}{L_i}\sum_{t=1}^{L_i}\left(\mathcal{L}_{i,t}^{\text{clip}} - \beta\mathbb{D}_{\text{KL}}\right)\right] \quad (1)$$

where $\mathcal{L}_{i,t}^{\text{clip}} = \min(\rho_{i,t}\hat{A}_{i,t}, \text{clip}(\rho_{i,t}, 1 \pm \epsilon)\hat{A}_{i,t})$ denotes the standard PPO surrogate loss. Here, the probability ratio is defined explicitly as $\rho_{i,t} = \frac{\pi_\theta(\mathbf{y}_{i,t}|q,\mathbf{y}_{i,<t})}{\pi_{\theta_{old}}(\mathbf{y}_{i,t}|q,\mathbf{y}_{i,<t})}$. The advantage $\hat{A}_{i,t}$ is computed using group statistics: $\hat{A}_{i,t} = (r_i - \mu_r)/\sigma_r$, where $\mu_r$ and $\sigma_r$ are the mean and standard deviation of the group rewards. The KL divergence is approximated via the estimator $\mathbb{D}_{\text{KL}} \approx \frac{\pi_{\text{ref}}}{\pi_\theta} - \log\frac{\pi_{\text{ref}}}{\pi_\theta} - 1$.

**GRPO for Visual Generation.** Recent works (Liu et al., 2025a; Xue et al., 2025) adapt this framework into Flow matching models for visual generation. Given the user request $q$, a group of image latents $\{\mathbf{z}^i\}_{i=1}^G$ are sampled. To enable the stochastic exploration required by GRPO, the deterministic flow is converted into a stochastic process via Euler-Maruyama discretization. The discrete update rule for the latent state $\mathbf{z}$ at each timestep $t$ is given by:

$$\mathbf{z}_{t-\Delta t} = \underbrace{\mathbf{z}_t + \left(v_\theta(\mathbf{z}_t) - \frac{\sigma_t^2}{2}\nabla\log p_t(\mathbf{z}_t)\right)\Delta t}_{\mu_\theta(\mathbf{z}_t)} + \sigma_t\sqrt{\Delta t}\epsilon \quad (2)$$

where $\sigma_t = a\sqrt{\frac{t}{1-t}}$, $a$ controls the noise level, and $\epsilon \sim \mathcal{N}(0, I)$ is standard Gaussian noise. This formulation explicitly defines the policy as a Gaussian distribution $\pi_\theta(\mathbf{z}_{t-\Delta t}|\mathbf{z}_t) = \mathcal{N}(\mu_\theta(\mathbf{z}_t), \sigma_t^2\Delta t I)$. Consequently, the log-probability for each step is computed analytically, and the probability ratio becomes the density ratio between the current and old policies: $\rho_t = \frac{\mathcal{N}(\mathbf{z}_{t-\Delta t};\mu_\theta, \Sigma_t)}{\mathcal{N}(\mathbf{z}_{t-\Delta t};\mu_{old}, \Sigma_t)}$. The objective sums over diffusion timesteps $T$ instead of tokens. Crucially, this formulation permits a closed-form KL divergence, calculated as the weighted $L_2$ distance between velocity fields:

$$\mathbb{D}_{\text{KL}}(\pi_\theta||\pi_{\text{ref}}) = w_t\|v_\theta(\mathbf{z}_t) - v_{\text{ref}}(\mathbf{z}_t)\|^2 \quad (3)$$

where the weighting term $w_t = \frac{\Delta t}{2}\left(\frac{\sigma_t(1-t)}{2t} + \frac{1}{\sigma_t}\right)^2$ is derived from the discretization parameters.

## 4. Methodology

This section details the core components of our method. We firstly introduce the AlphaGRPO algorithm in Sec. 4.1, followed by the design of the proposed Decompositional Verifiable Rewards, in Sec. 4.2. Lastly, Sec. 4.3 outlines the data curation process for constructing the training set.

### 4.1. AlphaGRPO

As shown in Figure 4, we propose **AlphaGRPO**, a unified framework that reinforces multimodal generation within an AR-Diffusion architecture. Next, we will introduce the details.

**Unified Trajectory Formulation.** We conceptualize the multimodal generation as a continuous generative process governed by a single unified model $\theta$. We define the output as a *hybrid trajectory* $\tau$ that concatenates the autoregressive reasoning sequence with the diffusion generation path for end-to-end joint optimization: $\tau = (\mathbf{y}, \mathbf{z}_1 \rightarrow \mathbf{z}_0)$. Specifically, the model first samples the discrete reasoning text tokens $\mathbf{y}$, which then serve as the conditional prior for the continuous visual trajectory $\{\mathbf{z}_t\}_{t=1}^0$.

This formulation unifies two distinct capabilities: (1) *Reasoning T2I*, where $\mathbf{y}$ acts as a cognitive bridge, planning spatial layouts and extracting specific world knowledge to ground the visual synthesis; and (2) *Self-Reflective Refinement*, where $\mathbf{y}$ diagnoses errors in previous outputs to guide refinement. Despite semantic differences, both tasks share the objective of maximizing visual quality conditioned on intermediate reasoning.

**Unified Optimization Objective.** The unified trajectory in the unified model allows us to employ GRPO to optimize the full trajectory $\tau$ end-to-end. For both tasks, the ultimate objective is to generate a high-quality image that gets higher rewards. By employing GRPO to optimize this multimodal generation problem, given context $\mathcal{C}$, we sample a group of $G$ trajectories $\{\tau_i\}_{i=1}^G$ where $\tau_i = (\mathbf{y}_i, \mathbf{z}_i)$. The reward $r(\mathbf{z}_i)$ is computed solely based on the final generated image $\mathbf{z}_i$ and the advantages $\hat{A}_i$ are obtained by normalizing the group reward $\{r_i\}$.

Crucially, since the reasoning $\mathbf{y}_i$ is the causal precursor to the image $\mathbf{z}_i$, we propagate the shared advantage $\hat{A}_i$ to update both policies. The unified objective is:

$$\mathcal{J}(\theta) = \mathbb{E}_{\pi_{old}}\left[\frac{1}{G}\sum_{i=1}^{G}\left(\lambda\mathcal{J}_{\text{AR}}^{(i)} + \mathcal{J}_{\text{Flow}}^{(i)}\right)\right], \quad (4)$$

where $\mathcal{J}_{\text{AR}}^{(i)}$, $\mathcal{J}_{\text{Flow}}^{(i)}$ and $\lambda$ represent the regularized PPO objectives for reasoning, generation and the balanced weight, respectively:

$$\mathcal{J}_{\text{AR}}^{(i)} = \mathcal{L}_{\text{AR}}(\mathbf{y}_i, \hat{A}_i) - \beta_{\text{AR}}\mathbb{D}_{\text{KL}}^{\text{AR}}, \quad (5)$$

$$\mathcal{J}_{\text{Flow}}^{(i)} = \mathcal{L}_{\text{Flow}}(\mathbf{z}_i, \hat{A}_i) - \beta_{\text{Flow}}\mathbb{D}_{\text{KL}}^{\text{Flow}}. \quad (6)$$

where $\beta_{\text{AR}}$ and $\beta_{\text{Flow}}$ are hyper-parameters. $\mathbb{D}_{\text{KL}}^{\text{AR}}$ and $\mathbb{D}_{\text{KL}}^{\text{Flow}}$ are KL divergences for reasoning and generation, respectively. Specifically, $\mathcal{L}_{\text{AR}}$ applies standard clipping to token probabilities, while $\mathcal{L}_{\text{Flow}}$ applies the same clipping strategy to the trajectory density ratios as detailed in Sec. 3.

**False-Positive Rectification.** In the self-reflective refine-

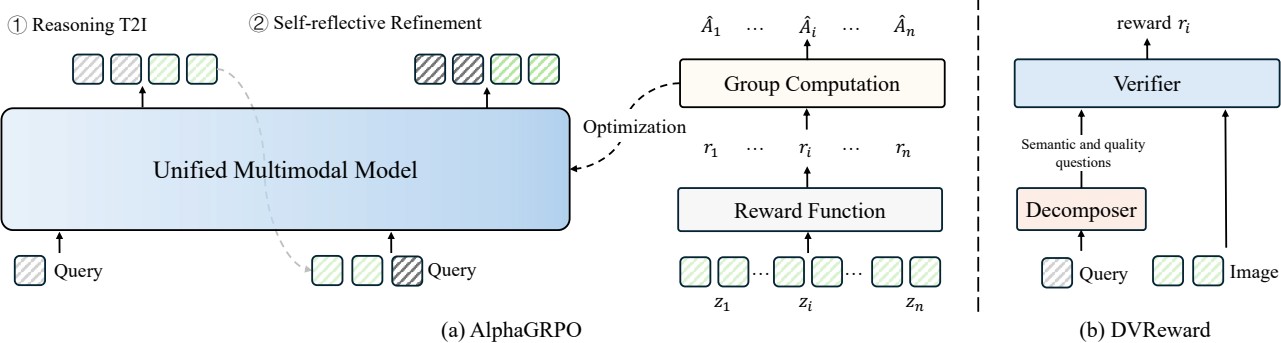

*Figure 4.* **An overview of the proposed framework.** (a) AlphaGRPO: The Unified Multimodal Model (UMM) is optimized using Group Reward Policy Optimization (GRPO). We optimize two tasks under the unified trajectories: (1) Reasoning T2I, which generates visual content from a query, and (2) Self-reflective Refinement, which improves upon previous outputs. (b) DVReward (The Decompositional Verifiable Reward) mechanism. To generate a robust reward signal, a Decomposer breaks down the initial query into specific semantic and quality questions/rubrics. A Verifier then assesses the generated image against these rubrics to produce a calibrated final reward. The gray box denotes the text token. The green box denotes the image token.

ment task, the optimization relies on the assumption that a valid trajectory must strictly improve upon the initial input. However, the group advantages calculated in GRPO can potentially assign a positive advantage to degraded refinement results that cause false-positive optimization. To eliminate this, we introduce False-Positive Rectification (FPR), which enforces a validity constraint by assigning the group minimum reward to the trajectories that fail to improve ($r(\mathbf{z}_i) \leq r(\mathbf{z}_{\text{init}})$). This operation guarantees that all ineffective refinement attempts result in negative advantages, strictly suppressing the likelihood of model degradation.

### 4.2. Decompositional Verifiable Reward

As in our pilot study (Sec. 2), holistic scalar rewards, i.e., VIEScore (Ku et al., 2024), suffer from uncalibrated quantification and poor discriminability. The arbitrary mapping from visual observations to scalar scores introduces inherent bias and noise, hindering effective GRPO training. To provide a robust reward signal for GRPO training on real-world multimodal generation, we introduce **Decompositional Verifiable Reward (DVReward)**, which replaces arbitrary holistic scoring with a calibrated verification process via request decomposition and confidence scoring.

**Request Decomposition.** Real-world user intents are multi-faceted and often under-specified. Current LLMs possess extensive world knowledge, enabling them to bridge the gap between abstract user intents and concrete visual evidence. Motivated by the Davidsonian Scene Graph (DSG) (Cho et al., 2023), we employ the LLM to decompose the user request $q$ into a comprehensive set of atomic, verifiable rubrics, covering semantic alignment and perceptual quality. Crucially, we enforce the LLM to perform physical visual grounding to convert abstract adjectives into observable physical phenomena. For example, instead of merely asking *"Is the coffee hot?"*, the model generates evidence-based questions like *"Is there steam rising from the cup?"*.

Specifically, we first generate the semantic rubrics $\mathcal{Q}_{\text{sem}}$ covering 10 dimensions, e.g., entity existence, attributes, and spatial relationships. Building upon these identified semantic anchors, we then generate the quality rubrics $\mathcal{Q}_{\text{qua}}$ covering 8 aspects, e.g., geometric completion, texture fidelity. Finally, a filtering process is applied to verify the evaluation validity of the generated rubrics.

**Confidence Scoring.** To assess the generated image $\mathbf{z}$, we employ the pre-trained MLLM, Qwen3VL-30B-A3B, as the verifier $\mathcal{V}$. For each rubric $s$, instead of discrete binary scores (Yes=1, No=0), which loses granularity, we utilize the probability ratio to extract the continuous confidence score. Let $P_{\text{Yes}}$ and $P_{\text{No}}$ denote the probability for the "Yes" and "No" token, respectively. The verification score $v_k \in [0, 1]$ is computed as $P_{\text{Yes}}/(P_{\text{Yes}} + P_{\text{No}})$. The final reward $r(\mathbf{z})$ is calculated as the geometric mean of the semantic scores $\bar{v}_{\text{sem}} = \frac{1}{|\mathcal{Q}_{\text{sem}}|} \sum_{s \in \mathcal{Q}_{\text{sem}}} \mathcal{V}(z, s)$ and quality scores $\bar{v}_{\text{qua}}$:

$$r(\mathbf{z}) = \sqrt{\bar{v}_{\text{sem}} \cdot \bar{v}_{\text{qua}}}. \tag{7}$$

### 4.3. Training Data Construction

To ensure the robustness and generalization of Alpha-GRPO, we curate a large-scale prompt set. We adopt a "Primitive-to-Prompt" bottom-up strategy to synthesize training data. First, we collect a visual elements pool containing a comprehensive pool of visual primitives (e.g., objects, attributes, spatial relations). Following the taxonomy of TIIF-Bench (Wei et al., 2025), we define 39 distinct compositional tasks, e.g., spatial reasoning, attribute binding, and counting. For each task, we employ the LLM, Qwen3-235B-A22B, to synthesize prompts by stochastically sampling elements from the pool. To ensure comprehensive complexity coverage, we instruct the model to generate prompts across three difficulty tiers (Easy, Medium, Hard). In total, we generate 19,500 training prompts (500 per task with a 3:5:2 difficulty ratio) and 1,024 test prompts.

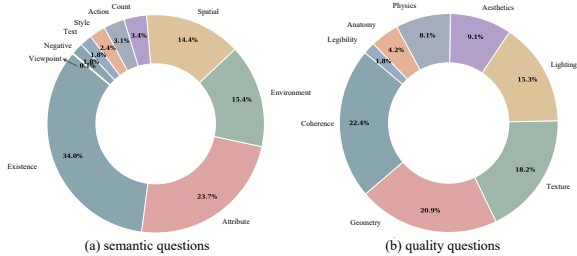

(a) semantic questions    (b) quality questions

*Figure 5.* **Distribution of rubric categories in the training set.**

We offline preprocess each prompt of the dataset to pre-generate the rubrics of DVReward. As illustrated in Figure 5, this process transforms raw text prompts into structured triplets $(q, \mathcal{Q}_{\text{sem}}, \mathcal{Q}_{\text{qua}})$. During AlphaGRPO training, we deploy Qwen3VL-30B-A3B (Bai et al., 2025) using SGLang (Zheng et al., 2024) as the verifier to verify the generated images. Although DVReward requires multiple MLLM inference passes to verify each sample, by asynchronously calling the reward model and optimizing the training procedure, the latency awaiting reward feedback can be reduced to a negligible level.

# 5. Experiments

## 5.1. Implementation Details

We implement AlphaGRPO on Bagel, a native Unified Multimodal Model. Unlike composite architectures that concatenate expert models, Bagel integrates understanding and generation within a single backbone trained on large-scale interleaved data, making it an ideal testbed for our Alpha-GRPO. To validate the versatility of our framework, we apply AlphaGRPO to two distinct tasks: Reasoning T2I and Self-Reflective Refinement. Unless otherwise specified, we adopt Self-Reflective Refinement as the default training setting for the main results.

**Training Settings.** For the reasoning T2I task, we use $T = 16$ sampling steps during training (convert the first 10 steps to SDE sampling for GRPO training) and $T = 40$ for evaluation. For self-reflective refinement, we employ 40 steps for both the initial generation and subsequent self-reflective refinement sampling. To encourage correction, the initial image is fixed as the sample with the lowest reward in the group. We stochastically sample 5 steps from the first 15 for GRPO training. We optimize 32 prompts for each training step. The group size is $G = 14$, noise level $a = 0.7$. Training image resolution is 512. The KL coefficients $\beta_{\text{AR}}$ and $\beta_{\text{Flow}}$ are both set to 0. $\lambda$ is 0.2.

## 5.2. Compare to State-of-the-Art

Rather than merely pursuing higher scores on the in-distribution test set, our primary objective is to assess the generalization capability of the model on standard downstream benchmarks after AlphaGRPO training. We evaluate

text-to-image performance on four comprehensive benchmarks: GenEval (Ghosh et al., 2023), TIIF-Bench (Wei et al., 2025), DPG-Bench (Hu et al., 2024), and WISE (Niu et al., 2025b). We compare AlphaGRPO against state-of-the-art generation-only models, including SD3 Medium (Esser et al., 2024) and FLUX.1 dev (Labs, 2024), as well as leading unified multimodal models such as Show-o (Xie et al., 2024), JanusPro (Chen et al., 2025b), and our backbone baseline BAGEL (Deng et al., 2025).

Table 1 demonstrates AlphaGRPO's consistent improvements across benchmarks. We highlight three observations: (1) **Training on low resolution boosts high-resolution performance.** Despite optimization at $512 \times 512$, Alpha-GRPO outperforms the Bagel baseline and generalizes effectively to higher resolutions. AlphaGRPO 1024px excels on TIIF-Bench and GenEval without high-res fine-tuning, verifying learned semantic alignment over pixel memorization. (2) **Training on self-reflective refinement generalizes to downstream tasks.** Rather than directly training on image generation tasks, training on self-reflective refinement enhances image generation performance comparable to that of directly optimizing the reasoning T2I task (e.g., matching 86.3 on DPG-Bench). This suggests AlphaGRPO instills generalized reasoning regardless of the training format. (3) **Inference time self-reflective refinement boosts performance.** By applying self-reflective refinement, the model obtains significant gains and achieves 4.5% and 3.4% improvement on TIIF-Bench short prompt and GenEval, respectively, outperforming Bagel by 5.8% and 2.9%.

**Multimodal Image Editing.** To further verify the effectiveness of AlphaGRPO, we evaluate it on GEdit-Bench, comparing it against open-source models Instruct-Pix2Pix (Isola et al., 2017), MagicBrush (Zhang et al., 2023), AnyEdit (Yu et al., 2025), OmniGen2 (Wu et al., 2025b), Step1X-Edit (Liu et al., 2025b), and BAGEL (Deng et al., 2025), as well as the closed-source models Gemini 2.0 (Gemini2, 2025) and GPT-4o (OpenAI, 2025). In Table 2, training on reasoning T2I can also improve performance on the editing task (+0.33). Crucially, training with self-reflective refinement further boosts the performance to an overall score of 7.08, achieving a gain of 0.52 over BAGEL. This indicates that the alignment learned via AlphaGRPO is not limited to generation but also enhances the model's ability to follow editing instructions and preserve visual consistency precisely.

## 5.3. Ablation Studies

In our ablation studies, we mainly evaluate the components on TIIF-Bench and GenEval at 512 resolution. All experiments are under the same setting.

**Effectiveness of DVReward.** We compare our DVReward to the human preference model, PickScore (Kirstain

*Table 1.* **Results on Text-to-Image benchmarks**. S and L denote Short and Long prompts, respectively. RT2I indicates the AlphaGRPO is trained on the Reasoning text-to-image generation task. **Bold** indicates the best performance. The results of BAGEL are reproduced.

| Model | TIIF Bench ↑ | | | | | | | | WISE ↑ | DPGBench ↑ | Geneval ↑ |
| | Basic | | Advanced | | Designer | | Overall | | | | |
| | S | L | S | L | S | L | S | L | Overall | Score | Score |
|---|---|---|---|---|---|---|---|---|---|---|---|
| *Generation Only Models* | | | | | | | | | | | |
| SD3 Medium (Esser et al., 2024) | 78.3 | 77.8 | 61.5 | 59.6 | 63.2 | 67.3 | 64.8 | 64.8 | 0.4 | 84.1 | 74.0 |
| FLUX.1 dev (Labs, 2024) | 83.1 | 78.7 | 65.8 | 68.5 | 70.7 | 71.5 | 71.1 | 71.8 | 0.5 | 83.8 | 82.0 |
| *Unified Multimodal Models* | | | | | | | | | | | |
| Show-o (Xie et al., 2024) | 73.1 | 75.8 | 55.0 | 50.9 | 53.7 | 50.4 | 59.7 | 58.9 | 0.35 | - | 69.0 |
| JanusPro (Chen et al., 2025b) | 79.3 | 78.3 | 59.7 | 58.8 | 65.8 | 60.3 | 66.5 | 65.0 | 0.35 | 84.2 | 80.0 |
| *Inference on 512 Resolution* | | | | | | | | | | | |
| BAGEL | 81.7 | 86.1 | 73.7 | 77.6 | 84.7 | 82.1 | 75.2 | 78.6 | - | 85.0 | 84.0 |
| **AlphaGRPO** (RT2I) | 85.5 | 84.2 | 77.4 | 78.9 | 84.3 | 86.6 | 78.9 | 79.5 | - | 86.0 | 85.0 |
| **AlphaGRPO** | 85.6 | 83.3 | 77.1 | 79.9 | 83.6 | 84.7 | 79.1 | 79.5 | - | 86.3 | 84.2 |
| + Inf. Self-Reflective Refinement | **89.8** | **88.8** | 88.2 | **88.7** | **84.7** | 83.2 | **83.9** | **83.2** | - | **87.9** | 88.2 |
| *Inference on 1024 Resolution* | | | | | | | | | | | |
| BAGEL | 83.4 | 83.7 | 75.2 | 76.7 | 79.8 | 73.5 | 76.4 | 76.2 | 0.53/0.70 | 85.1 | 86.6 |
| **AlphaGRPO** (RT2I) | 84.8 | 85.9 | 79.9 | 78.4 | 80.2 | 80.2 | 78.7 | 79.5 | 0.50/0.69 | 85.9 | 87.4 |
| **AlphaGRPO** | 85.4 | 82.9 | 75.6 | 77.5 | 81.7 | 84.3 | 77.7 | 78.1 | **0.53/0.71** | 85.1 | 86.1 |
| + Inf. Self-Reflective Refinement | 89.4 | 88.4 | **88.4** | 84.2 | 79.9 | **84.0** | 82.2 | 81.4 | - | 86.6 | **89.5** |

et al., 2023) and MLLM-based reward models, HPSv3 (Ma et al., 2025) and UnifiedReward (Wang et al., 2025b), and holistic scalar scoring method, i.e., VIEScore (with the same MLLM backbone) on two base models, Stable Diffusion 3.5 Medium (SD3.5M) under FlowGRPO setting (Liu et al., 2025a) and Bagel under AlphaGRPO setting. As shown in Table 3, VIEScore would degrade SD3.5M on TIIF-Bench on long prompts and degrade Bagel on GenEval. UnifiedReward and HPSv3 both degrade BAGEL on TIIF-Long and GenEval. In contrast, DVRewardconsistently improves all benchmarks and surpasses VIEScore and PickScore as reward models.

**Impact of Confidence Scoring.** We compare the confidence scoring in DVRewardagainst a hard Binary Score baseline (which outputs only 0 or 1 score). As shown in Table 4, the Confidence Score achieves a clear advantage on TIIF-Bench Long prompts (79.5 vs. 78.9) and boosts the overall GenEval score to 85.1, surpassing the Binary Score (84.0). Unlike the binary score that treats "barely correct" and "perfectly correct" identically, the confidence score preserves the model's uncertainty, offering a smoother gradient landscape for optimization.

**Effect of False Positive Rectification (FPR).** As shown in Table 5, enabling FPR yields consistent gains across all metrics, most notably boosting performance on the challenging TIIF-Bench Long split from 77.8 to 79.5. This confirms that filtering out spurious success signals is critical for preventing the model from overfitting to noisy rewards and ensuring

*Table 2.* **Comparison on GEdit-Bench-EN.** All metrics are higher-is-better (↑) and evaluated by GPT-4.1.

| Type | Model | GEdit-Bench ↑ | | |
| | | G_SC | G_PQ | G_O |
|---|---|---|---|---|
| *Private* | Gemini 2.0 (Gemini2, 2025) | 6.73 | 6.61 | 6.32 |
| | GPT-4o (OpenAI, 2025) | **7.85** | **7.62** | **7.53** |
| *Open-source* | Instruct-Pix2Pix (Isola et al., 2017) | 3.58 | 5.49 | 3.68 |
| | MagicBrush (Zhang et al., 2023) | 4.68 | 5.66 | 4.52 |
| | AnyEdit (Yu et al., 2025) | 3.18 | 5.82 | 3.21 |
| | OmniGen2 (Wu et al., 2025b) | 7.16 | 6.77 | 6.41 |
| | Step1X-Edit (Liu et al., 2025b) | 7.09 | 6.76 | 6.70 |
| | BAGEL (Deng et al., 2025) | 7.36 | 6.83 | 6.52 |
| *Ours* | **AlphaGRPO** (RT2I) | 7.54 | 7.18 | 6.85 |
| | **AlphaGRPO** | **7.67** | **7.46** | **7.08** |

robust semantic alignment.

**Impact of Composite Rubric Types.** We validate the effectiveness of the two reward aspects in DVReward. As shown in Table 6, integrating perceptual quality rubrics alongside Semantic Alignment yields consistent improvements on both TIIF-Bench and GenEval. This indicates that enforcing perceptual quality constraints prevents the model from generating "semantically correct but visually degraded" samples, thereby ensuring high-fidelity generation.

# 6. Related Work

**Unified Multimodal Models (UMMs).** The landscape of UMMs has evolved from pure autoregressive architectures to advanced hybrid paradigms. Early works (Team, 2024; Wu et al., 2024; Wang et al., 2024) adopted a Pure AR

*Table 3*. **Comparison of reward models.** We compare our DVReward to the human preference reward, PickScore, and holistic scalar score method, VIEScore.

| Model | Reward | TIIF ↑ | | Geneval ↑ |
|-------|--------|--------|------|-----------|
| | | Short | Long | Overall |
| SD3.5M | - | 74.0 | 73.2 | 79.8 |
| SD3.5M | PickScore | 77.6 | 76.3 | 80.4 |
| SD3.5M | VIEScore | 76.2 | 72.9 | 82.9 |
| SD3.5M | DVReward | **79.1** | **77.7** | **86.0** |
| BAGEL | - | 75.2 | 78.6 | 84.0 |
| BAGEL | HPSv3 | 78.5 | 77.1 | 83.4 |
| BAGEL | UnifiedReward | **79.2** | 77.3 | 83.7 |
| BAGEL | VIEScore | 79.1 | 77.9 | 81.7 |
| BAGEL | DVReward | 78.9 | **79.5** | **85.1** |

*Table 4*. **Ablation of the confidence scoring in DVReward.** Training on reasoning T2I task.

| Module | TIIF ↑ | | GenEval ↑ |
|--------|--------|------|-----------|
| | Short | Long | Overall |
| Binary Score | **79.0** | 78.9 | 84.0 |
| Confidence Score | 78.9 | **79.5** | **85.1** |

*Table 5*. **Ablation of the False Positive Rectification**. Training on the self-reflective refinement task.

| FPR | TIIF ↑ | | GenEval ↑ |
|-----|--------|------|-----------|
| | Short | Long | Overall |
| ✓ | 79.1 | 79.5 | 84.2 |
| ✗ | 77.9 | 77.8 | 83.7 |

*Table 6*. **Ablation of composited rubric type in DVReward.** Training on the reasoning T2I task, evaluated on 1024 resolution.

| Semantic | Quality | TIIF ↑ | | GenEval ↑ |
|----------|---------|--------|------|-----------|
| | | Short | Long | Overall |
| ✓ | ✗ | 76.3 | 77.7 | 87.3 |
| ✓ | ✓ | 78.7 | 79.5 | 87.4 |

approach, tokenizing images into discrete codes to model vision and language in a unified sequence. To overcome the generation quality limitations of discrete tokens, the field shifted towards the AR-Diffusion paradigm. Initial attempts (Jin et al., 2023; Wang et al., 2025a) employed diffusion decoders primarily for upscaling or reconstruction. This evolution eventually bifurcated into two distinct streams: (1) *Composite UMMs* (Tong et al., 2025; Pan et al., 2025; Chen et al., 2025a; Lin et al., 2025; Li et al., 2025) utilize connectors to bridge specialized LLMs and DiTs; and (2) *Native UMMs* (Liao et al., 2025; Deng et al., 2025) integrate understanding and continuous generation into a single backbone. In this work, we focus on the Native UMMs to explore joint optimization of text and image generation via unified GRPO training.

**Reinforcement Learning for Multimodal Generation.** Reinforcement Learning (RL) (Schulman et al., 2017; Shao et al., 2024) has proven highly effective in enhancing the reasoning and perception capabilities of LLMs and MLLMs. Recently, this success has extended to visual generation (Liu et al., 2025a; Xue et al., 2025), optimizing diffusion models for specific objectives. However, the application of RL to UMMs remains underexplored. Previous studies have explored the unified RL on pure AR unified models (Mao et al., 2025; Nie et al., 2025), while they relied on task-specific reward designs. To date, there is a lack of systematic exploration regarding the GRPO training of AR-Diffusion-based UMMs. In this paper, we bridge this gap by conducting the first comprehensive study on unified GRPO training

for AR-Diffusion UMMs, covering reasoning T2I and self-reflective refinement tasks, and verifying its effectiveness and generalization on downstream benchmarks.

**Reward Models for Multimodal Generation.** The efficacy of RL hinges on the quality of reward signals. Early multimodal rewards (Kirstain et al., 2023; Wu et al., 2023; Xu et al., 2023) primarily relied on fine-tuning CLIP encoders or training regressors on human preference data. Recent MLLM-based approaches (Wang et al., 2025b; Xu et al., 2024; Xiong et al., 2025; Jin et al., 2025) have shifted toward SFT on large-scale preference datasets to achieve fine-grained alignment. However, this potentially compromises the MLLM's generalist capabilities by forcing it into a specialized domain. Conversely, VIEScore (Ku et al., 2024) directly instructs the MLLMs to give scalar scores but often suffers from output instability. In this work, we explore a new reward method, DVReward, leveraging the inherent understanding ability of MLLMs to construct robust, verifiable rewards without sacrificing their general capabilities.

## 7. Conclusion

In this paper, we present AlphaGRPO, a novel reinforcement learning framework designed to unlock the latent reasoning and self-refinement capabilities of AR-Diffusion-based unified multimodal models. To provide a reliable reward signal for multimodal generation in real-world scenarios, we propose Decompositional Verifiable Reward (DVReward) powered by an open-source MLLM to decompose the user request into atomic, verifiable questions/rubrics. We validate the effectiveness of AlphaGRPO on reasoning text-to-image and self-reflective refinement tasks. Experimental results demonstrate that AlphaGRPO with DVReward significantly generalizes and enhances performance in downstream T2I benchmarks and Editing benchmarks without an additional cold-start stage. These findings underscore the potential of reinforcing unified models' intrinsic primitives to achieve robust, generalized multimodal intelligence.

## Acknowledgements

This work is supported by the National Natural Science Foundation of China (No. 62422606) and Hong Kong Research Grant Council General Research Fund (No. 17213925).

## Impact Statement

This paper presents work aimed at leveraging Reinforcement Learning to unlock and enhance the inherent capabilities of Unified Multimodal Models. We acknowledge that the alignment process, guided by specific reward models, carries the potential risk of amplifying existing biases present in the reward signals. However, since both our backbone models and reward models are derived from publicly available open-source checkpoints, our method does not introduce new categories of societal risks or ethical concerns beyond those already inherent in the base models themselves.

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

# A. Appendix

## A.1. Limitations and Future Work

The base model, BAGEL, occasionally generates noisy or blurry artifacts at a resolution of 512 and outputs unexpected reasoning patterns during self-reflection, which potentially compromises the overall effectiveness of AlphaGRPO. To address this without introducing external knowledge, future work could employ Reinforcement Fine-Tuning (RFT). By sampling high-quality data consistent with the model's intrinsic distribution for fine-tuning, we aim to improve the stability of AlphaGRPO training.

Regarding the self-reflective refinement task, our current approach relies primarily on outcome rewards, calculating the reward based solely on the final output using DVReward (i.e., text-to-image alignment). We have not yet explored process rewards. Future work includes incorporating consistency rewards to ensure semantic fidelity before and after refinement, as well as providing intermediate supervision to verify whether the self-reflection successfully identifies specific mistakes.

## A.2. Detailed Implementation

**Training Setting.** We apply LoRA ($r = 32, \alpha = 64$) to all linear layers in the attention and MLP modules. Training is conducted on 64 NVIDIA A100 GPUs. On each 8-GPU node, we dedicate 7 GPUs to training and 1 GPU to serving the reward model (Qwen3VL-30B-A3B) to calculate DVReward. We only train 380 steps with update minibatch size equals the rollout batch size. We do not employ a cold start phase before AlphaGRPO training. We incorporate a format penalty for the thinking text during training, assigning a value of -0.5 for incorrect tags and 0 for correct ones. This penalty is directly added to the image reward to compute the advantage for the entire trajectory. We utilize classifier-free guidance (CFG) (Ho & Salimans, 2022) during training; specifically, the reasoning text-to-image task uses a text CFG of 4.0, while the self-reflective refinement task employs both a text CFG of 4.0 and an image CFG of 2.0. Regarding sampling for the thinking text, we use a temperature of 1.0 to maintain exploration and a top-p value of 0.8 to mitigate the generation of garbled text. Finally, to ensure stability during self-reflective refinement training, we utilize single-turn conversations rather than multi-turn conversations (containing the history of the initially generated images).

**Training Data Construction.** We directly drop those prompts that have more than 50 questions/rubrics. We visualize the distribution of the training set we used in Figure 7. For each rubric's type, we put the definitions in Table 16 and Table 17. To better understand the decomposer, we provide qualitative examples from two angles. Table 7 presents

| Prompt | BAGEL | AlphaGRPO (RT2I) |

On the right, a rabbit sits on a stump, while **another rabbit remains off the stump.**

A woman is **not browsing wooden books** on a wooden surface.

**Beside the vibrant and dazzling spherical gemstone**, …, stands a simple yet functional plastic cylindrical salt shaker,…

**Beneath a crisp, cerulean sky, a spirited girl**, her cheeks kissed with the rosy hue of winter's chill, exuberantly partakes in a lively snowball fight, as the **serene snowman, standing vigil on the right**, remains a steadfast witness and silent guardian of the frosty frolic.

*Figure 6.* **Qualitative comparison of AlphaGRPO (RT2I) and BAGEL.** RT2I means reasoning text-to-image generation.

one representative source prompt and the corresponding rubrics for each category, illustrating the diversity of rubric types our decomposer can produce. Table 8 provides end-to-end prompt-to-rubrics decompositions, where a single input prompt is decomposed into both semantic rubrics that verify prompt-grounded content and quality rubrics that probe visual fidelity.

## A.3. Additional Experimental Results

**Effect of Rubric Count.** According to the Fig. 7, we do not ask the decomposer to generate limited rubrics and it would generate several tens of rubrics that would increase the effort of verification. To validate whether fewer rubrics are sufficient, we merge the full rubric set into at most 10 core rubrics per sample using Qwen3-235B-A22B. Experimental results in Table 9 show that the reduced-rubric variant is competitive on TIIF-Bench but drops substantially on GenEval, especially in Spatial and Count. This supports the need for fine-grained atomic verification: merging multiple constraints into coarse rubrics weakens the reward signal for individual relations and attributes.

**Evaluation with Additional Reward Metrics.** We further evaluate 1K real-user prompts, sampled from Pick-a-Pic, using AestheticScore, CLIPScore (Hessel et al., 2021), and PickScore (Kirstain et al., 2023), none of which are used as the training reward. AlphaGRPO consistently improves over

*Table 7.* **Qualitative examples of generated questions grouped by category.** For each category, we show one source prompt and one representative generated question. Semantic questions evaluate whether prompt conditions are satisfied, while quality questions evaluate visual fidelity and rendering quality.

| Category | Source prompt | Generated question |
|---|---|---|
| **Semantic questions** | | |
| Existence | Three camels and six clouds. | Are there camels? |
| Attribute | A blue vial is not glowing. | Is the vial blue? |
| Environment | A red fox moves across the snow. | Is there snow? |
| Spatial | A platform is above a bag. | Is the platform above the bag? |
| Count | Three camels and six clouds. | Are there six clouds? |
| Action | A red squirrel holds a bottle. | Is the squirrel holding the bottle? |
| Style | A cat in a painterly style with visible brush-strokes. | Is the image in a painterly style? |
| Text | A melon is in front of a sign that says LONDON. | Does the sign say LONDON? |
| Negative | A blue vial is not glowing. | Is the vial not glowing? |
| Viewpoint | A realistic close-up of a journal page on a wooden desk with a hand-drawn dinosaur skeleton and handwritten notes. | Is this a close-up view? |
| **Quality questions** | | |
| Geometry | The pentagon is behind the circle. | Is the pentagon geometrically accurate with five clearly defined straight sides and correct angular proportions? |
| Coherence | Two energy bursts, one larger than the other. | Are the two energy bursts clearly separated and individually discernible without merging or unintended blending? |
| Texture | A cat sits on a couch. | Is the cat's fur texture detailed and visually consistent with real feline fur? |
| Lighting | A blue vial is not glowing. | Does the vial lack emissive lighting effects, confirming it is not glowing? |
| Aesthetics | A chrome-plated katana swings on the right side of the frame. | Does the pose or trajectory of the katana suggest dynamic movement consistent with a swing? |
| Physics | A cat sits on a couch. | Is the contact between the cat and the couch consistent with the physics of a seated object? |
| Anatomy | A butterfly floats to the left. | Does the butterfly have anatomically plausible wings and body structure consistent with a typical butterfly? |
| Legibility | Three shrimp on the left of a framed quote. | Is the text within the frame legible, with clear characters and no gibberish or visual corruption? |

BAGEL under all three metrics, showing that the observed gains are not tied only to the DVReward verifier.

**Human Evaluation and Alignment with DVReward.** We conduct human evaluation on 200 real-user prompts sampled from Pick-a-Pic. Three annotators compare each BAGEL and AlphaGRPO pair on overall preference, prompt following, and perceptual quality. As shown in Table 11, human annotators prefer AlphaGRPO more often than BAGEL in all three dimensions, confirming stronger prompt adherence without perceptual-quality degradation. And we also use DVRewardto evaluate the pairwise preference between BAGEL and AlphaGRPO, the results are shown in the last row of Table 11. DVReward also selects AlphaGRPO more often than BAGEL at the pairwise level, suggesting that the DVReward training signal is aligned with human preference.

**Compare to Inference-Time SRR on BAGEL.** We con-

duct the inference-time self-reflective refinement on Bagel and compare the improvement with the AlphaGRPO w/ Inf. SRR to validate the effectiveness of RL training. Applying inference-time SRR to the zero-shot BAGEL baseline yields limited gains (+2.3, -0.1, and +2.3 on TIIF-S, TIIF-L, and GenEval), whereas AlphaGRPO + Inf. SRR produces substantially larger gains (+4.8, +3.7, and +4.0). This indicates that AlphaGRPO improves the model's self-reflection behavior rather than relying solely on the inference-time refinement procedure.

**KL Coefficient Ablation.** We ablate KL regularization and find that adding $\beta_{AR} = \beta_{Flow} = 0.001$ does not provide consistent gains. It improves TIIF-S slightly but degrades TIIF-L and GenEval, so we set both KL coefficients to 0 in the default setting.

*Table 8.* **Example prompt-to-question decompositions.** Given an input prompt, the decomposer in DVRewardproduces semantic questions for prompt coverage and quality questions for visual fidelity.

| Source prompt | Generated semantic questions | Generated quality questions |
|---|---|---|
| A blue ship moves through the sky. | 1. Is there a ship?
2. Is the ship blue?
3. Is the ship moving through the sky? | 1. Is the ship geometrically coherent with a structurally plausible hull and super-structure?
2. Is the ship clearly defined and visually distinct from the surrounding sky without fusion or blending artifacts?
3. Does the ship exhibit a consistent blue coloration across its surface with realistic material variation, such as paint sheen or weathering?
4. Does the ship's orientation and motion cues, such as implied speed or wake effects, suggest dynamic movement through the air?
5. Is the lighting on the ship consistent with the surrounding sky environment, supporting its presence in aerial space? |
| A blue toilet is on the left. A glowing cluster is on the right. | 1. Is there a toilet?
2. Is there a glowing cluster?
3. Is the toilet blue?
4. Is the toilet on the left?
5. Is the glowing cluster on the right? | 1. Is the toilet geometrically accurate with a properly formed bowl, tank, and base structure?
2. Is the toilet clearly separated from its surroundings without visual fusion or distortion artifacts?
3. Does the glowing cluster emit a consistent and visually distinct luminosity that suggests an internal light source?
4. Is the cluster composed of coherent, non-melting elements that maintain structural integrity?
5. Is the blue color of the toilet consistent across its surface with realistic material shading and hue fidelity? |

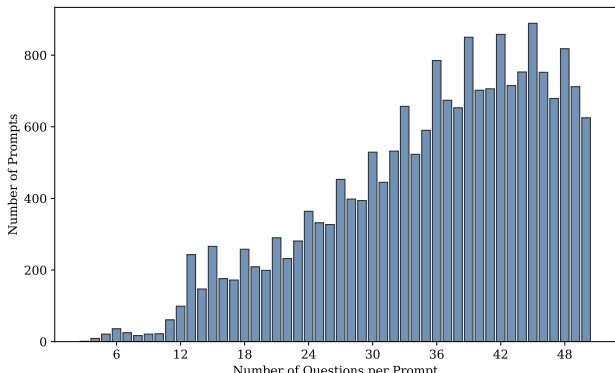

*Figure 7.* **Distribution of the rubrics numbers in the synthesized prompt.**

*Table 9.* **Ablation on the number of verification rubrics.** "-" denotes the BAGEL baseline without AlphaGRPO training.

| # Rubrics | TIIF | | GenEval | | |
|---|---|---|---|---|---|
| | Short | Long | Overall | Spatial | Count |
| - | 75.2 | 78.6 | 84.0 | 71.0 | 74.0 |
| 10 | 78.2 | 79.7 | 81.1 | 63.0 | 71.8 |
| All | 78.9 | 79.5 | 85.0 | 74.3 | 82.5 |

*Table 10.* Evaluation on 1K real-user prompts with additional reward metrics.

| Metric | BAGEL | AlphaGRPO |
|---|---|---|
| AestheticScore | 5.9583 | 5.9624 |
| CLIPScore | 0.8984 | 0.9102 |
| PickScore | 0.8208 | 0.8246 |

## A.4. Analysis of Efficiency of DVReward

Because DVReward uses an external MLLM verifier, naive reward calls can block rollout and training, reducing GPU utilization. We therefore combine a high-performance serving engine (SGLang) with decentralized reward serving and asynchronous scheduling, so that online verification is overlapped with rollout and policy updates. The resulting reward-waiting bubble time is reduced from $40.8$ s to $9.72 \times 10^{-6}$ s.

The efficiency comes from the following design choices:

- Multiple verification calls for the same sample share the same system prompt and image prefix, allowing KV-cache reuse without recomputing image features or repeated LLM prefill.

- Each question adds only a small verification cost: about 20 prefill tokens and one decoded Yes/No token. All questions for a sample are processed in parallel on the reward server.

- The verifier, Qwen3VL-30B-A3B, activates only about 3B parameters and supports high throughput (over 4K prefill tokens/s and more than 500 running requests on one A100), resulting in low online verification latency.

- Decentralized reward servers remove the cross-node communication bottleneck in multi-node training. Instead of sending all image requests to a centralized server, we deploy one reward server on each 8-GPU node and keep reward requests local.

*Table 11.* **Human Evaluation and the pairwise alignment between DVRewardand human preference.**

| Evaluator | Evaluation | BAGEL Win | Tie | AlphaGRPO Win |
|---|---|---|---|---|
| | Overall | 30.5% | 29.0% | 40.5% |
| Human | Prompt Following | 25.5% | 38.0% | 36.5% |
| | Perceptual Quality | 39.0% | 18.0% | 43.0% |
| DVReward | Overall | 34.7% | 14.1% | 51.3% |

*Table 12.* Comparison with applying inference-time SRR to BAGEL.

| Model | TIIF-S | TIIF-L | GenEval |
|---|---|---|---|
| BAGEL | 75.2 | 78.6 | 84.0 |
| BAGEL + Inf. SRR | 77.47 | 78.45 | 86.3 |
| AlphaGRPO | 79.1 | 79.5 | 84.2 |
| AlphaGRPO + Inf. SRR | 83.9 | 83.2 | 88.2 |

- Asynchronous scheduling eliminates most idle time. During RL training, a batch is split into four mini-batches for sequential rollout. Each mini-batch is submitted to the reward server immediately after roll-out, while reward collection is delayed until all mini-batches finish and before the corresponding policy update. This overlaps reward computation with later roll-outs and the previous mini-batch update.

## A.5. Detailed Benchmark Results

In this section, we provide the detailed evaluation results of each benchmark. Please note that most of the time, we could not accurately reproduce the officially reported baseline (BAGEL) performance. To make a fair comparison, we report and compare with the value we reproduce. Table 18 shows that AlphaGRPO trained on the reasoning T2I task could improve the performance on counting and position, while applying the inference self-reflective refinement could significantly improve the color attri. and counting performance.

Table 19 presents the detailed breakdown of TIIF-Bench (Wei et al., 2025), highlighting that Alpha-GRPO achieves remarkable gains in challenging fine-grained metrics, such as reaching a perfect 100.0 score on Style consistency with inference-time refinement. Besides, our approach significantly improves semantic precision, boosting Text rendering performance by over 13 points (53.85 vs. 40.72) and consistently outperforming the baseline in the complex Advanced Following subset, validating that unified alignment effectively resolves semantic binding failures.

Table 20 shows the detailed results of DPG-Bench (Hu et al., 2024). Our AlphaGRPO, particularly when equipped with inference-time self-reflective refinement, achieves a state-of-the-art Overall score of **87.86**. This performance significantly improves upon the Bagel baseline (85.07). Notably, the high scores in *Attribute* (91.91) and *Relation* (91.67)

*Table 13.* **Ablation of KL coefficients.**

| $\beta_{AR}$ | $\beta_{Flow}$ | TIIF-S | TIIF-L | GenEval |
|---|---|---|---|---|
| 0 | 0 | 78.9 | 79.5 | 85.0 |
| 0.001 | 0.001 | 79.2 | 78.7 | 84.1 |

*Table 14.* **Ablation of reward-serving optimizations.**

| Decentralized Serving | Asynchronous Scheduling | Bubble Time (s) |
|---|---|---|
| No | No | $40.8 \pm 3.42$ |
| Yes | No | $7.28 \pm 0.73$ |
| Yes | Yes | $9.72 \times 10^{-6} \pm 3.63 \times 10^{-7}$ |

categories confirm that our unified alignment strategy effectively enhances the model's ability to handle dense, complex prompts with precise semantic binding.

Table 21 presents the evaluation on the WISE benchmark (Niu et al., 2025b), which assesses world knowledge reasoning. Notably, AlphaGRPO demonstrates substantial gains in scientific domains requiring precise structural understanding. Most significantly, in the *Chemistry* category, our method boosts the performance of the Self-CoT setting to 0.64, outperforming the Bagel (0.58). This indicates that our unified alignment strategy effectively enhances the model's ability to reason about and generate complex scientific concepts.

## A.6. More Visualization

To provide a more comprehensive assessment of our proposed framework, we present extensive qualitative comparisons below.

**Reasoning Text-to-Image Generation.** Figure 6 presents a side-by-side comparison between AlphaGRPO (trained on Reasoning T2I) and the BAGEL baseline. The visual results highlight our model's superior capability in handling complex compositional prompts. For instance, in the first row, AlphaGRPO precisely executes the spatial instruction to place one rabbit "off the stump," whereas the baseline incorrectly positions both rabbits on stumps. Similarly, in the third row, our model correctly distinguishes diverse objects (a gemstone and a salt shaker), avoiding the object hallucination (generating two spheres) observed in the baseline. These examples confirm that AlphaGRPO effectively mitigates semantic mismatches and enforces strict adherence to spatial and logical constraints.

**Efficacy of Self-Reflective Refinement.** Figure 8 visualizes the progressive improvement brought by our Inference-time Self-Reflective Refinement (Inf. SRR). Comparison across columns reveals the model's capability to diagnose and repair specific generation failures. For instance, in the first row, while the initial generation suffers from attribute leakage (dressing both figures in yellow), the refined output

*Table 15.* **Reward-waiting bubble time comparison.** PickScore is invoked synchronously, while DVReward uses our decentralized serving with asynchronous scheduling; the gap reflects scheduling overhead rather than raw reward compute cost.

| Reward Model | Bubble Time (s) |
|---|---|
| PickScore | $4.19 \times 10^{-2} \pm 1.29 \times 10^{-3}$ |
| DVReward | $9.72 \times 10^{-6} \pm 3.63 \times 10^{-7}$ |

*Table 16.* **Definitions of Semantic Alignment Question Type.**

| Category | Description |
|---|---|
| Style | Artistic medium (sketch, oil, photo), visual genre (anime, cyberpunk), or image format. |
| Environment | Background setting, weather, time of day, lighting atmosphere, or location context. |
| Viewpoint | Camera parameters: angle (top-down), shot size (close-up), lens type (fisheye), or framing. |
| Existence | Presence or visibility of specific subjects/objects (Binary Yes/No), excluding quantity. |
| Count | Numerical quantity or multiplicity of specific objects (e.g., 'three', 'a pair', 'single'). |
| Attribute | Static visual properties of objects: color, material, shape, texture, size, or attire. |
| Action | Dynamic movements (running), physical activities, body poses (sitting), or active states. |
| Spatial | Relative positioning (left/right, behind), depth relations, or interactions like holding/wearing. |
| Text | Presence, spelling, or visibility of specific written words, characters, signage, or logos. |
| Negative | Explicit absence of elements or verification that something is NOT present. |

*Table 17.* **Definitions of Perceptual Quality Question Type.**

| Aspect | Description |
|---|---|
| Geometry | Structural integrity, perspective logic, and shape correctness for inorganic objects (buildings, cars). |
| Anatomy | Biological correctness of humans/animals: limb proportions, hands, faces, and skeletal logic. |
| Texture | Realism of surface materials, fine details, resolution, noise levels, and material fidelity. |
| Coherence | Object integrity: no unintended melting, fusion, detachment, or illogical blending. |
| Lighting | Consistency of illumination, shadow direction, light source logic, and reflections. |
| Physics | Physical plausibility: gravity (ground contact/floating), motion blur, and fluid dynamics. |
| Legibility | Readability of text: spelling accuracy, clear glyphs, and lack of gibberish. |
| Aesthetics | Overall visual appeal, adherence to art style, and freedom from digital artifacts/glitches. |

successfully corrects the smaller person's attire to a "different color." Similarly, in the bottom row, the refinement step resolves the object fusion issue, transforming the merged

texture into two distinct, side-by-side pizzas. These examples validate that our test-time scaling strategy effectively rectifies semantic ambiguities that the one-pass generation fails to resolve.

**Generalization to Image Editing.** Figure 9 showcases the robust generalization of AlphaGRPO on the GEdit benchmark (Liu et al., 2025b). Despite not being explicitly trained for editing, our model outperforms the baseline in maintaining visual consistency and content integrity. A striking example is observed in the second row ("Change to a white background"): while the Bagel baseline catastrophically degrades the photorealistic box into a line drawing (sketch), AlphaGRPO successfully modifies the background while strictly preserving the object's original texture and geometry. Similarly, in the first row, our model accurately synthesizes realistic "snow" coverage, whereas the baseline merely desaturates the scene, failing to capture the textural changes required by the instruction.

*Table 18.* **Evaluation of text-to-image generation ability on GenEval benchmark.** 'Gen. Only' stands for an image generation model, and 'Unified' denotes a model that has both understanding and generation capabilities. † refers to the methods using LLM rewriter. Our model's results and Bagel all use LLM rewriter.

| Model | Single Obj. | Two Obj. | Counting | Colors | Position | Color Attri. | Overall↑ |
|---|---|---|---|---|---|---|---|
| *Gen. Only Models* | | | | | | | |
| PixArt-$\alpha$ (Chen et al., 2024) | 98.0 | 50.0 | 44.0 | 80.0 | 8.0 | 7.0 | 48.0 |
| Emu3-Gen (Wang et al., 2024) | 98.0 | 71.0 | 34.0 | 81.0 | 17.0 | 21.0 | 54.0 |
| SDXL (Podell et al., 2024) | 98.0 | 74.0 | 39.0 | 85.0 | 15.0 | 23.0 | 55.0 |
| DALL-E 3 (Betker et al., 2023) | 96.0 | 87.0 | 47.0 | 83.0 | 43.0 | 45.0 | 67.0 |
| SD3-Medium (Esser et al., 2024) | 99.0 | 94.0 | 72.0 | 89.0 | 33.0 | 60.0 | 74.0 |
| FLUX.1-dev† (Labs, 2024) | 98.0 | 93.0 | 75.0 | 93.0 | 68.0 | 65.0 | 82.0 |
| *Unified Models* | | | | | | | |
| SEED-X (Ge et al., 2024) | 97.0 | 58.0 | 26.0 | 80.0 | 19.0 | 14.0 | 49.0 |
| TokenFlow-XL (Qu et al., 2025) | 95.0 | 60.0 | 41.0 | 81.0 | 16.0 | 24.0 | 55.0 |
| ILLUME (Wang et al., 2025a) | 99.0 | 86.0 | 45.0 | 71.0 | 39.0 | 28.0 | 61.0 |
| Transfusion (Zhou et al., 2024) | - | - | - | - | - | - | 63.0 |
| Emu3-Gen† (Wang et al., 2024) | 99.0 | 81.0 | 42.0 | 80.0 | 49.0 | 45.0 | 66.0 |
| Show-o (Xie et al., 2024) | 98.0 | 80.0 | 66.0 | 84.0 | 31.0 | 50.0 | 68.0 |
| Janus-Pro-7B (Chen et al., 2025b) | 99.0 | 89.0 | 59.0 | 90.0 | 79.0 | 66.0 | 80.0 |
| MetaQuery-XL† (Pan et al., 2025) | - | - | - | - | - | - | 80.0 |
| *Inference on 512 resolution* | | | | | | | |
| Bagel | 99.1 | 95.0 | 74.1 | 90.4 | 71.0 | 74.8 | 84.0 |
| **AlphaGRPO† (RT2I)** | 98.1 | 95.7 | 82.5 | 91.0 | 74.3 | 68.8 | 85.1 |
| **AlphaGRPO†** | 98.8 | 97.0 | 75.6 | 91.0 | 69.8 | 73.3 | 84.2 |
| + Self-reflective Refinement | 99.7 | 97.7 | 83.4 | 93.1 | 73.8 | 81.5 | 88.2 |
| *Inference on 1024 resolution* | | | | | | | |
| Bagel | 98.4 | 94.7 | 81.3 | 94.7 | 74.0 | 76.3 | 86.6 |
| **AlphaGRPO (RT2I)** | 98.8 | 95.2 | 82.8 | 93.6 | 76.3 | 77.8 | 87.4 |
| **AlphaGRPO** | 99.1 | 96.0 | 80.3 | 94.7 | 71.0 | 75.5 | 86.1 |
| + Self-reflective Refinement | 99.4 | 97.2 | 87.2 | 97.1 | 72.3 | 83.8 | 89.5 |

*Table 19.* Performance of closed-source models and state-of-the-art open-source models on TIIF-Bench **testmini** subset. Evaluated systems are grouped into (i) diffusion-based open-source models, (ii)autoregressive open-source models, and (iii) closed-source models. The results of AlphaGRPO and BAGEL are evaluated by gpt4.1. "Inf. SRR" indicates executing the inference-time self-reflective refinement.

| Model | Overall | | Basic Following | | | | | | | | Advanced Following | | | | | | | | | | | | Designer | |
| | | | Avg | | Attribute | | Relation | | Reasoning | | Avg | | Attribute +Relation | | Attribute +Reasoning | | Relation +Reasoning | | Style | | Text | | Real World | |
| | short | long | short | long | short | long | short | long | short | long | short | long | short | long | short | long | short | long | short | long | short | long | short | long |
| Diffusion based Open-Source Models | | | | | | | | | | | | | | | | | | | | | | | | |
| FLUX.1 dev | 71.09 | 71.78 | 83.12 | 78.65 | 87.05 | 83.17 | 87.25 | 80.39 | 75.01 | 72.39 | 65.79 | 68.54 | 67.07 | 73.69 | 73.84 | 73.34 | 69.09 | 71.59 | 66.67 | 66.67 | 43.83 | 52.83 | 70.72 | 71.47 |
| SD XL | 54.96 | 42.13 | 65.72 | 53.28 | 59.33 | 50.83 | 77.57 | 62.57 | 60.32 | 46.57 | 49.73 | 36.22 | 47.82 | 35.57 | 56.22 | 45.34 | 52.59 | 36.09 | 73.33 | 60.00 | 16.83 | 0.83 | 50.92 | 41.59 |
| SD 3 | 67.46 | 66.09 | 78.32 | 77.75 | 83.33 | 79.83 | 82.07 | 78.82 | 71.07 | 74.07 | 61.46 | 59.56 | 61.07 | 64.07 | 68.84 | 70.34 | 50.96 | 57.84 | 66.67 | 76.67 | 59.83 | 20.83 | 63.23 | 67.34 |
| SD 3.5 L | 71.15 | 66.96 | 78.34 | 79.56 | 79.50 | 76.50 | 80.96 | 83.21 | 72.46 | 78.71 | 67.67 | 61.18 | 66.46 | 61.89 | 73.53 | 74.15 | 60.03 | 61.53 | 73.33 | 63.33 | 70.52 | 42.52 | 64.43 | 66.39 |
| AR based Open-Source Models | | | | | | | | | | | | | | | | | | | | | | | | |
| Llamagen | 41.67 | 38.22 | 53.00 | 50.00 | 48.33 | 42.33 | 59.57 | 60.32 | 51.07 | 47.32 | 35.89 | 32.61 | 38.82 | 31.57 | 40.84 | 47.22 | 49.59 | 46.22 | 46.67 | 33.33 | 0.00 | 0.00 | 39.73 | 35.62 |
| Show-o | 59.72 | 58.86 | 73.08 | 75.83 | 74.83 | 79.83 | 78.82 | 78.32 | 65.57 | 69.32 | 53.67 | 50.38 | 60.95 | 56.82 | 68.59 | 68.96 | 66.46 | 56.22 | 63.33 | 66.67 | 3.83 | 2.83 | 55.02 | 50.92 |
| Infinity | 62.07 | 62.32 | 73.08 | 75.41 | 74.33 | 76.83 | 72.82 | 77.57 | 72.07 | 71.82 | 56.64 | 54.98 | 60.44 | 55.57 | 74.22 | 64.71 | 60.22 | 59.71 | 80.00 | 73.33 | 10.83 | 23.83 | 54.28 | 56.89 |
| JanusPro | 66.50 | 65.02 | 79.33 | 78.25 | 79.33 | 82.33 | 78.32 | 73.32 | 80.32 | 79.07 | 59.71 | 58.82 | 66.07 | 56.20 | 70.46 | 70.84 | 67.22 | 59.97 | 60.00 | 70.00 | 28.83 | 33.83 | 65.84 | 60.25 |
| *Inference on 512 resolution* | | | | | | | | | | | | | | | | | | | | | | | | |
| BAGEL | 75.21 | 78.56 | 81.73 | 86.11 | 85.50 | 88.00 | 84.99 | 85.39 | 74.69 | 84.94 | 73.66 | 77.61 | 77.68 | 81.55 | 67.77 | 76.48 | 78.58 | 77.86 | 90.00 | 90.00 | 33.03 | 40.72 | 84.70 | 82.09 |
| **AlphaGRPO (RT2I)** | 78.92 | 79.48 | 85.46 | 84.15 | 88.50 | 85.50 | 88.12 | 86.56 | 79.77 | 80.38 | 77.41 | 78.85 | 81.05 | 82.77 | 74.38 | 80.52 | 79.30 | 75.83 | 90.00 | 83.33 | 44.80 | 53.85 | 84.33 | 86.57 |
| **AlphaGRPO** | 79.05 | 79.50 | 85.56 | 83.32 | 89.50 | 85.50 | 85.34 | 83.60 | 81.86 | 80.86 | 77.12 | 79.87 | 78.59 | 83.95 | 71.81 | 78.94 | 83.02 | 79.36 | 86.67 | 93.33 | 51.13 | 45.25 | 83.58 | 84.70 |
| + Inf. SRR | 83.86 | 83.20 | 89.77 | 88.75 | 91.50 | 88.00 | 88.70 | 88.12 | 89.10 | 90.14 | 88.23 | 88.73 | 82.53 | 84.31 | 82.15 | 85.07 | 85.80 | 85.48 | 100.00 | 100.00 | 51.13 | 45.25 | 84.70 | 83.21 |
| *Inference on 1024 resolution* | | | | | | | | | | | | | | | | | | | | | | | | |
| BAGEL | 76.42 | 76.15 | 83.44 | 83.72 | 87.50 | 89.00 | 86.03 | 84.35 | 76.77 | 77.81 | 75.16 | 76.68 | 79.34 | 83.12 | 70.38 | 74.58 | 78.36 | 75.58 | 93.33 | 86.67 | 36.20 | 40.72 | 79.85 | 73.51 |
| **AlphaGRPO (RT2I)** | 78.70 | 79.48 | 84.83 | 85.92 | 89.00 | 89.50 | 88.81 | 88.36 | 76.69 | 79.89 | 78.42 | 79.21 | 79.38 | 84.09 | 77.48 | 77.37 | 81.05 | 78.84 | 90.00 | 90.00 | 45.70 | 47.06 | 80.22 | 80.22 |
| **AlphaGRPO** | 77.74 | 78.09 | 85.39 | 82.90 | 89.00 | 89.50 | 87.31 | 82.96 | 79.85 | 76.25 | 75.62 | 77.49 | 79.46 | 83.32 | 73.28 | 76.13 | 76.48 | 75.56 | 90.00 | 90.00 | 42.53 | 44.80 | 81.72 | 84.33 |
| + Inf. SRR | 82.24 | 81.39 | 89.36 | 88.38 | 91.00 | 91.00 | 87.36 | 84.99 | 89.71 | 89.14 | 88.41 | 84.18 | 82.65 | 83.58 | 89.88 | 81.78 | 81.36 | 80.50 | 96.67 | 93.33 | 42.53 | 44.80 | 79.85 | 83.96 |

*Table 20.* Evaluation of text-to-image generation ability on DPG-Bench (Hu et al., 2024) benchmark. * is our reproduced results.

| Method | Global↑ | Entity↑ | Attribute↑ | Relation↑ | Other↑ | Overall↑ |
|---|---|---|---|---|---|---|
| *Gen. Only Models* | | | | | | |
| Hunyuan-DiT (Li et al., 2024) | 84.59 | 80.59 | 88.01 | 74.36 | 86.41 | 78.87 |
| PixArt-Σ (Chen et al., 2024) | 86.89 | 82.89 | 88.94 | 86.59 | 87.68 | 80.54 |
| DALLE3 (Betker et al., 2023) | 90.97 | 89.61 | 88.39 | 90.58 | 89.83 | 83.50 |
| SD3-medium (Esser et al., 2024) | 87.90 | 91.01 | 88.83 | 80.70 | 88.68 | 84.08 |
| FLUX.1-dev (Labs, 2024) | 82.10 | 89.50 | 88.70 | 91.10 | 89.40 | 84.00 |
| OmniGen (Xiao et al., 2025) | 87.90 | 88.97 | 88.47 | 87.95 | 83.56 | 81.16 |
| *Unified Multimodal Models* | | | | | | |
| Show-o (Xie et al., 2024) | 79.33 | 75.44 | 78.02 | 84.45 | 60.80 | 67.27 |
| EMU3 (Wang et al., 2024) | 85.21 | 86.68 | 86.84 | 90.22 | 83.15 | 80.60 |
| TokenFlow-XL (Qu et al., 2025) | 78.72 | 79.22 | 81.29 | 85.22 | 71.20 | 73.38 |
| Janus (Wu et al., 2025a) | 82.33 | 87.38 | 87.70 | 85.46 | 86.41 | 79.68 |
| Janus Pro (Chen et al., 2025b) | 86.90 | 88.90 | 89.40 | 89.32 | 89.48 | 84.19 |
| BLIP3-o 4B (Chen et al., 2025a) | - | - | - | - | - | 79.36 |
| BLIP3-o 8B (Chen et al., 2025a) | - | - | - | - | - | 81.60 |
| BAGEL (Deng et al., 2025) | 88.94 | 90.37 | 91.29 | 90.82 | 88.67 | 85.07 |
| UniWorld-V1 (Lin et al., 2025) | 83.64 | 88.39 | 88.44 | 89.27 | 87.22 | 81.38 |
| OmniGen2 (Wu et al., 2025b) | 88.81 | 88.83 | 90.18 | 89.37 | 90.27 | 83.57 |
| *Inference on 512 resolution* | | | | | | |
| BAGEL* | 88.94 | 90.37 | 91.29 | 90.82 | 88.67 | 85.07 |
| **AlphaGRPO (RT2I)** | 89.99 | 92.20 | 88.49 | 90.89 | 89.12 | 85.98 |
| **AlphaGRPO** | 84.99 | 90.93 | 91.22 | 92.51 | 90.11 | 86.25 |
| + Self-reflective Refinement | 85.50 | 90.79 | 91.91 | 91.67 | 93.51 | 87.86 |
| *Inference on 1024 resolution* | | | | | | |
| BAGEL* | 87.42 | 92.46 | 90.75 | 91.92 | 84.96 | 85.17 |
| **AlphaGRPO (RT2I)** | 87.42 | 92.46 | 90.75 | 91.92 | 84.96 | 85.87 |
| **AlphaGRPO** | 89.21 | 89.43 | 90.20 | 92.26 | 90.39 | 85.08 |
| + Self-reflective Refinement | 86.38 | 91.97 | 90.76 | 92.77 | 89.99 | 86.62 |

*Table 21.* **Comparison of world knowledge reasoning on WISE.** WISE examines the complex semantic understanding and world knowledge for T2I generation. 'Gen. Only' stands for an image generation model, and 'Unified' denotes a model that has both understanding and generation capabilities.

| Type | Model | Cultural | Time | Space | Biology | Physics | Chemistry | Overall↑ |
|------|-------|----------|------|-------|---------|---------|-----------|----------|
| *Gen. Only* | SDXL (Podell et al., 2024) | 0.43 | 0.48 | 0.47 | 0.44 | 0.45 | 0.27 | 0.43 |
| | SD3.5-large (Esser et al., 2024) | 0.44 | 0.50 | 0.58 | 0.44 | 0.52 | 0.31 | 0.46 |
| | PixArt-Alpha (Chen et al., 2024) | 0.45 | 0.50 | 0.48 | 0.49 | 0.56 | 0.34 | 0.47 |
| | FLUX.1-dev (Labs, 2024) | 0.48 | 0.58 | 0.62 | 0.42 | 0.51 | 0.35 | 0.50 |
| *Unified* | Janus (Wu et al., 2025a) | 0.16 | 0.26 | 0.35 | 0.28 | 0.30 | 0.14 | 0.23 |
| | VILA-U (Wu et al., 2024) | 0.26 | 0.33 | 0.37 | 0.35 | 0.39 | 0.23 | 0.31 |
| | Show-o-512 (Xie et al., 2024) | 0.28 | 0.40 | 0.48 | 0.30 | 0.46 | 0.30 | 0.35 |
| | Janus-Pro-7B (Chen et al., 2025b) | 0.30 | 0.37 | 0.49 | 0.36 | 0.42 | 0.26 | 0.35 |
| | Emu3 (Wang et al., 2024) | 0.34 | 0.45 | 0.48 | 0.41 | 0.45 | 0.27 | 0.39 |
| | MetaQuery-XL (Pan et al., 2025) | 0.56 | 0.55 | 0.62 | 0.49 | 0.63 | 0.41 | 0.55 |
| | BAGEL (Deng et al., 2025) | 0.44 | 0.55 | 0.68 | 0.44 | 0.60 | 0.39 | 0.52 |
| | BAGEL w/ Self-CoT (Deng et al., 2025) | 0.76 | 0.69 | 0.75 | 0.65 | 0.75 | 0.58 | 0.70 |
| | AlphaGRPO | 0.44 | 0.55 | 0.64 | 0.46 | 0.62 | 0.46 | 0.53 |
| | AlphaGRPO w/ Self-CoT | 0.75 | 0.70 | 0.74 | 0.66 | 0.77 | 0.64 | 0.71 |

| Prompt | BAGEL | AlphaGRPO | AlphaGRPO + Inf. SRR |
|---|---|---|---|

A larger person wearing yellow clothing stands next to a smaller person dressed in a different color

a detailed oil painting that captures the essence of an elderly raccoon adorned with a distinguished black top hat. The raccoon's fur is depicted with textured, swirling strokes reminiscent of Van Gogh's signature style, and it clutches a bright red apple in its paws. The background swirls with vibrant colors, giving the impression of movement around the still figure of the raccoon.

A realistic scene depicts a brown oven and a purple train positioned side by side without overlap. The oven is placed on the left side of the frame. It has a deep, rich brown color with a matte finish, and its metallic handle reflects subtle light. The oven's surface is smooth, with visible seams and a small vent on top. On the right side, a purple train stands prominently. The train's body is a vibrant, glossy purple, with silver accents along its edges and wheels. The train's windows are dark and reflective, adding depth to its design.

There are two pizzas placed side by side on a wooden table. Each pizza has a perfectly round shape and a thin, crispy crust. The crust is golden brown, with slight charring around the edges. The surface of both pizzas is evenly topped with melted mozzarella cheese, which appears bubbly and slightly browned in certain spots. Visible slices of pepperoni are scattered across each pizza in an even pattern. The pizzas are identical in size and toppings, with no noticeable differences between them. The lighting highlights their textures, creating a photographic and lifelike representation of the scene.

*Figure 8.* **Qualitative comparison of AlphaGRPO and BAGEL.** "Inf. SRR" indicates using inference-time self-reflective refinement to improve the previous results.

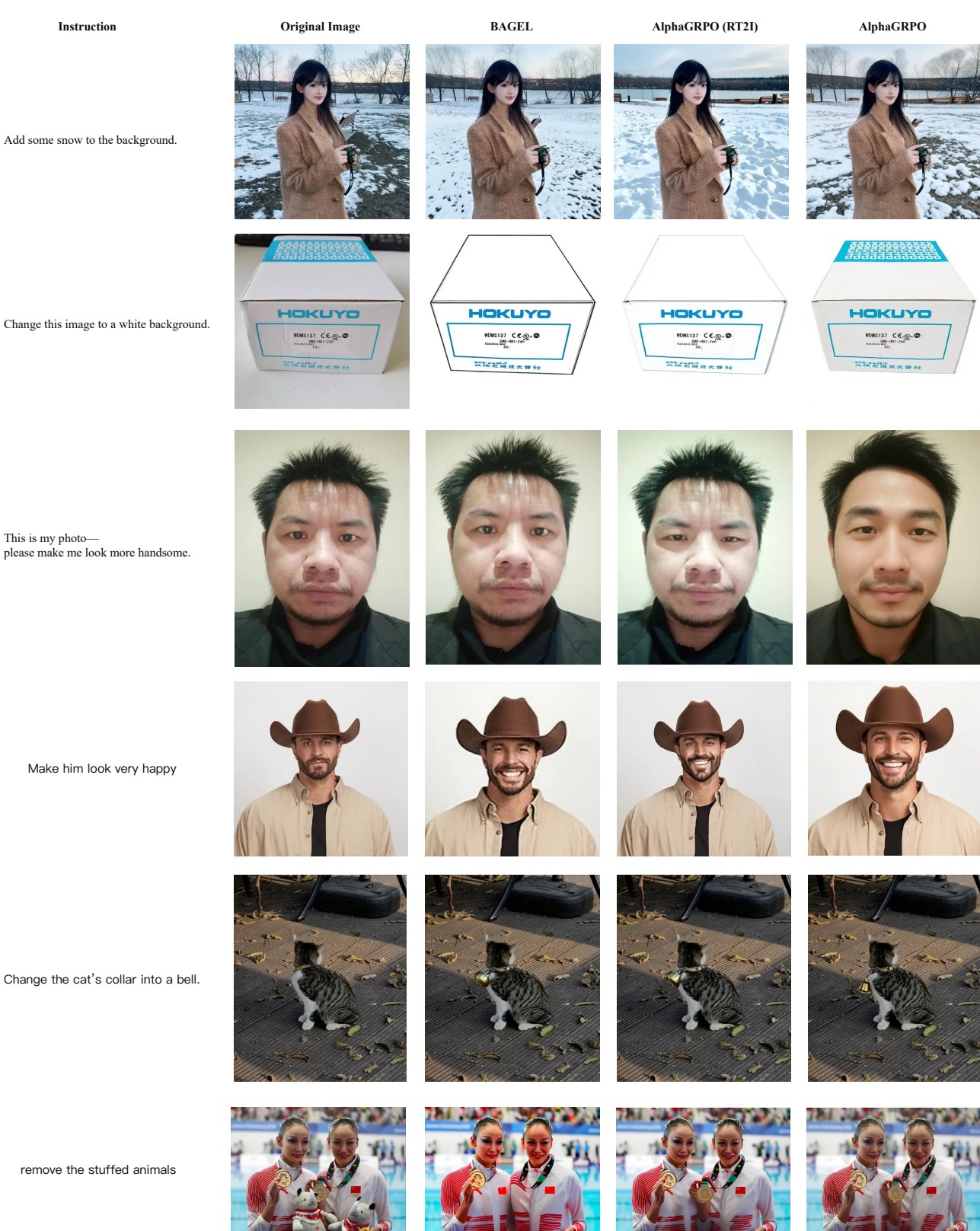

*Figure 9.* **Qualitative results of editing benchmark, GEdit (Liu et al., 2025b)**

