# OpenReview forum: "AlphaGRPO: Unlocking Self-Reflective Multimodal Generation in Unified Multimodal Models via Decompositional Verifiable Reward"
_ICML.cc/2026/Conference — ICML 2026 regular_

### Official Review · Reviewer_EYyk · 2026-02-21

**Soundness:** 3
**Presentation:** 3
**Significance:** 2
**Originality:** 2
**Overall Recommendation:** 4
**Confidence:** 3

**Summary:**

The paper proposes AlphaGRPO, an RL framework that jointly optimizes both the text and image modalities of a UMM using GRPO. Within a unified trajectory formulation, it optimizes two tasks: Reasoning T2I, which generates visual content from a query, and Self-Reflective Refinement (SRR), which iteratively improves previous outputs. The paper reports that additional SRR steps further boost performance by reducing misalignment in the initial generations.

The authors also introduce DVReward, which reformulates each prompt into a set of semantic and quality questions to compute stable, verifiable reward signals, aiming to preserve the model’s underlying understanding while improving reward stability. The method is evaluated on Reasoning T2I and SRR, showing consistent gains over the BAGEL baseline. Extensive experiments on GenEval, TIIF-Bench, DPG-Bench, WISE, and GEdit, together with ablations, support the claimed benefits of DVReward.

**Compliance With Llm Reviewing Policy:**

Affirmed.

**Final Justification:**

The authors' dedicated rebuttal mostly addressed my major concerns. So I raised my score.

**Key Questions For Authors:**

## Questions
1) In Table 3, what exactly does the human preference reward represent, and how is it computed?
2) In Figure 1, what does the pi plot look like when applying the inference-time SRR step to the zero-shot BAGEL model? More specifically, does AlphaGRPO + SRR yield a larger performance gain than zero-shot + SRR?
3) In Section 5.1, why are both $\beta_{AR}$ and $\beta_{Flow}$ set to 0? Are there any ablation results supporting this choice? This setting does not appear to follow the original GRPO configuration.
4) In Tables 9–11, under 512 and 1024 resolution settings, why is the performance ordering between the RT2I model and the naive AlphaGRPO model not consistent?
5) What is Table 11 intended to demonstrate? Also, why do the duplicated rows have different values?

## Additional Comments
1) The current version of Section 4.3 does not feel very reader-friendly. I recommend strengthening the explanation with an illustrative figure and/or more concrete, step-by-step descriptions.
2) It would be helpful to include more qualitative examples for each category of the generated questions to improve interpretability.

**Limitations:**

The authors should provide more qualitative examples of failure cases.

**Strengths And Weaknesses:**

# Strengths
1) Rigorous and broad benchmarking across multiple public evaluation benchmarks (TIIF, GenEval, DPG, WISE) and an editing benchmark (GEdit) with gains reported at multiple resolutions and with/without inference-time SRR.

2) Ablations isolating DVReward vs. VIEScore/PickScore, impact of confidence scoring versus binary scoring, FPR on/off, and inclusion of quality questions provide useful evidence of where improvements originate.

3) The method is clearly structured: pilot study → preliminaries (GRPO for text and diffusion) → unified objective → reward design → data and experiments. Figures effectively illustrate the pipeline and qualitative outcomes.

4) The decompositional QA approach yields simple yet effective gains for online RL of UMMs.

# Weaknesses
1) The description in Section 4.2 (confidence scoring) does not sufficiently verify the quality of the generated questions. In particular, if the feedback is restricted to binary Yes/No signals, how are ambiguous edge cases handled where the question cannot be answered decisively?  The reviewer recommends additional analysis on whether using a three-way, verifiable reward (e.g., Yes / No / Cannot Decide) improves performance and stability.

2) The claim of being the “first” to bring GRPO to AR–Diffusion UMMs appears overstated, as contemporaneous works applying GRPO/FlowGRPO to AR/diffusion pipelines with Reason–Reflect–Refine stages are not clearly discussed. The paper should position its novelty more carefully relative to these works [1,2], and the reviewer recommends that the authors expand and deepen the related-work discussion in Section 3.

3) Why did the authors choose not to formulate the questions as multiple-choice question answering (MCQA) pairs? Compared to binary Yes/No feedback, additional comparative analysis is needed to assess the performance benefits of using MCQA-style questions.

4) Human evaluation is lacking. To make the results more trustworthy, the paper should include additional human (or human-level) evaluations assessing: (1) how well the generated images follow the prompts, (2) whether question-based VLM judgments align with human judgments on the same samples, and (3) whether image faithfulness degrades compared to the BAGEL baseline. Human-level evaluations using strong proprietary models (e.g., Gemini-3.0 or GPT-5.2) would also be helpful.

5) In Table 12, BAGEL and AlphaGRPO achieve very similar performance (approximately 0.52 vs. 0.53), and even with self-CoT the results remain nearly identical (around 0.70 vs. 0.71). The paper should provide additional clarification about the WISE benchmark and include analysis explaining why the performance gap is so small.

[1] Mao et al., UniRL: Self-Improving Unified Multimodal Models via Supervised and Reinforcement Learning, arxiv 2025.
[2] Jiang et al., Co-Reinforcement Learning for Unified Multimodal Understanding and Generation, NeurIPS 2025.

---

> ### Author Rebuttal · Authors · 2026-03-31
>
> We thank the reviewer for the thorough evaluation and for recognizing the rigorous benchmarking, ablation design, and the effectiveness of DVReward.
>
> &nbsp;
>
> ### **W1: 3-way scoring (Yes/No/Cannot Decide)**
> ---
> We clarify that DVReward **does not use hard binary (0 or 1) answers but a confidence score** P(Yes)/(P(Yes)+P(No)) in [0,1]. It inherently captures ambiguity, like a score near 0.5 functionally equivalent to "Cannot Decide."
>
> As requested, we conducted the experiment on 3-way scoring, computing confidence as P(Yes)/(P(Yes)+P(No)+P(Ambiguous)). The results show no meaningful difference. Table 4 of the main paper further confirms that confidence scoring outperforms hard binary scoring.
> |Scoring|TIIF-S|TIIF-L|GenEval|
> |:-|:-:|:-:|:-:|
> |**2-way (ours)**|78.92|79.48|85.05|
> |3-way|78.86|79.52|85.05|
>
> &nbsp;
>
> ### **W2: Extend related works**
> ---
> We'll expand the related work in Section 6. We note that UniRL and Co-RL both operate on **discrete-token UMMs** (VQ-based AR or discrete diffusion generation), which face fundamentally different RL challenges from continuous flow matching. To our knowledge, **AlphaGRPO is the first to apply GRPO to native AR-Continuous Diffusion UMMs**, with high potential to extend to interleaved multimodal generation.
>
> Besides, **AlphaGRPO does not require SFT to distill external knowledge** and proposes a general reward model, DVReward, to provide robust reward signals for general multimodal generation scenarios. AlphaGRPO, training on RT2I and SRR, **both demonstrating strong generalization to OOD downstream benchmarks**.
>
> &nbsp;
> ### **W3: MCQA**
> ---
> Constructing MCQA would require generating plausible distractors per question, which (1) significantly **increases design complexity** and (2) **collapses continuous confidence in DVReward into discrete choices**, losing fine-grained reward signal. For this reason, we choose Yes/No scoring, which has proved effective in Table 4.
>
> &nbsp;
> ### **W4: Human evaluation**
> ---
> We conduct human evaluation on 200 real-user prompts (sampled from Pick-a-Pic), please see our response to Reviewer a6Kf W3/Q2 for the full table. It addresses all three points:
> - (1) Prompt following: AlphaGRPO wins 36.5% vs BAGEL 25.5%, confirming stronger prompt adherence.
> - (2) Image faithfulness: Perceptual quality shows AlphaGRPO 43.0% vs BAGEL 39.0%, confirming no degradation.
> - (3) VLM-human alignment (see below): **Both DVReward and human evaluation consistently prefer AlphaGRPO**, with aligned preference directions:
> |Evaluator|BAGEL Win|Tie|AlphaGRPO Win|
> |:-|:-:|:-:|:-:|
> |DVReward|34.7%|14.1%|**51.3%**|
> |Human|30.5%|29.0%|**40.5%**|
>
> &nbsp;
> ### **W5: Performance on WISE**
> ---
> Our training prompts focus on compositional generation tasks and **do not cover world knowledge** (which WISE evaluates). However, AlphaGRPO not only maintains performance on WISE but even shows **significant gains on Chemistry (0.64 vs 0.58)**, confirming that **AlphaGRPO training could generalize to OOD downstream benchmarks**. Beyond WISE, it also shows consistent improvements on other OOD benchmarks, e.g., +3.7 on TIIF-S, +1.3 on DPG-Bench, and +0.56 on GEdit.
>
> &nbsp;
> ### **Q1: "Human preference reward" in Table 3**
> ---
> "Human preference reward" in Table 3 refers to PickScore, which is trained on the human preference dataset (Pick-a-Pic). We'll clarify this in the revision.
>
> &nbsp;
> ### **Q2: Compare to BAGEL+SRR**
> ---
> Applying inference-time SRR to BAGEL yields limited gains (+2.3/−0.1/+2.3 on  TIIF-S/L/GenEval), while AlphaGRPO + Inf. SRR achieves ~2× larger gain (+4.8/+3.7/+4.0), demonstrating that **AlphaGRPO improves the SRR ability and learns more effective reflection patterns during RL training**.
> |Model|TIIF-S|TIIF-L|GenEval|
> |:-|:-:|:-:|:-:|
> |BAGEL|75.2|78.6|84.0|
> |BAGEL+Inf.SRR|77.47|78.45|86.3|
> |AlphaGRPO|79.1|79.5|84.2|
> |AlphaGRPO+Inf.SRR|**83.9**|**83.2**|**88.2**|
>
> &nbsp;
> ### **Q3: Choice of β_AR and β_Flow**
> ---
> We find that using KL regularization (β > 0) does not provide consistent gains. See our response to Reviewer 8UXc W5 for the discussion and ablation.
>
> &nbsp;
> ### **Q4: Tables 9–11 performance ordering inconsistency**
> ---
> "AlphaGRPO" without (RT2I) denotes training on Self-Reflective Refinement (SRR). The ordering difference is expected because RT2I and SRR optimize different objectives (RT2I targets reasoning-driven image generation, while SRR focuses on reflection on error and refinement). More importantly, **SRR is not trained on any T2I task**, yet **SRR consistently improves over the BAGEL baseline across all T2I benchmarks and Editing benchmark**, demonstrating strong task generalization.
>
> &nbsp;
> ### **Q5,6,7: Improve Readability**
> ---
> - In Table 11, the first three rows are evaluated on 512 resolution and the below three are 1024 resolution.
>
> - We'll add a step-by-step figure for the DVReward pipeline and a supplementary table with example questions from each of the 10 semantic and 8 quality categories.
>
> We'll update these in the revision.

---

> > ### Author Rebuttal · Reviewer_EYyk · 2026-04-01
> >
> > **Reviewer's Feedback on Rebuttal**
> >
> > I would like to thank the authors for their thorough and dedicated responses. Most of my primary concerns have been addressed effectively, and I appreciate the additional experimental results provided.
> >
> > **[Updates on Previous Concerns]**
> >
> > Tables 9–11 Inconsistency: Thank you for the clarification regarding the performance ordering in Tables 9–11. I now understand that RT2I and SRR optimize different objectives and that the generalization of the SRR-trained model to T2I tasks is a notable result.
> >
> > OOD Generalization: My concerns regarding the model’s ability to generalize to OOD downstream benchmarks have been resolved by your explanation and the additional evidence provided.
> >
> > Scoring Mechanism: I understood the advantages of the Confidence Score over hard binary or MCQA-style feedback. The reviewer acknowledged that my initial concern regarding this point was based on a misunderstanding of the scoring implementation, and the 3-way scoring experiment further solidified this.
> >
> > **[Strengths of the Rebuttal]**
> >
> > Human Evaluation (W4): The inclusion of human evaluation on 200 real-user prompts is highly convincing and demonstrates a clear alignment between DVReward and human judgment.
> >
> > Comparison with BAGEL+SRR (Q2): The data showing that AlphaGRPO achieves significantly larger gains than simply applying inference-time SRR to the baseline effectively justifies the necessity of the proposed RL framework.
> >
> >
> > **[Suggestions for Final Revision]**
> >
> > Related Work & Technical Distinction: Regarding the authors' claim that "AlphaGRPO is the first to apply GRPO to native AR-Continuous Diffusion UMMs," please ensure the Camera-Ready version includes a more detailed discussion in the Related Works section.
> >
> > Readability: Please incorporate the qualitative figures and more failure cases faithfully integrated into the final manuscript.
> >
> > **Conclusion**
> >
> > I am impressed by the authors' thoroughness and the quality of the rebuttal. The additional data significantly strengthens the paper’s claims. In light of these clarifications and the authors' dedicated efforts, I am raising my score to 4.

---

> > > ### Author Response · Authors · 2026-04-02
> > >
> > > Thank you for your positive and constructive feedback. We are glad that most of your concerns have been well addressed.
> > >
> > > We promise to expand the Related Work section with a more in-depth discussion. We will also carefully revise the manuscript to improve readability and faithfully incorporate additional qualitative figures and failure cases in the final version.
> > >
> > > We also kindly note that the score in the system may not yet reflect the update mentioned in your response. we would appreciate it if you could update it at your convenience.
> > >
> > > Once again, we sincerely appreciate the time and effort you have dedicated throughout the review process. Your feedback has been invaluable in strengthening our work.

---

### Official Review · Reviewer_a3xb · 2026-03-11

**Soundness:** 2
**Presentation:** 4
**Significance:** 3
**Originality:** 2
**Overall Recommendation:** 4
**Confidence:** 5

**Summary:**

This paper proposes AlphaGRPO, a RL framework for unified understanding and generation models. It focuses on T2I tasks with reasoning component and iterative self-refinement via reflection. It proposes Decompositional Verifiable Reward, which decomposes a T2I task into multiple verifiable atomic statements and use the understanding capabilities of MLLM to verify these statements. The final results is aggregated into the reward score.

**Compliance With Llm Reviewing Policy:**

Affirmed.

**Final Justification:**

Concerns are fully resolved, especially regarding the latency. Hence, I'm increasing the score

**Key Questions For Authors:**

Main Questions
1. Please clarify how is "asynchronously scheduling" implemented and what is the overhead in terms of the compute capacity requried to serve the 30B reward model.
2. How does DVReward compared with other VLM based reward like HPSv3 and UnifiedReward?

Overall, I'm impressed by the novelty of this paper and its strong experiment results. However, this paper need more efforts in showing that the performance gain is not simply achieved by scaling to a 30B VLM reward model, which is less interesting. I am willing to increase my score if author offers comparison with HPSv3 and UnifiedReward, or provide any other alternative experiment results highlighting the effectiveness of the "decomposition" pipeline (e.g. reducing the number of questions used in computing the reward and see if performance drops)

**Limitations:**

Yes

**Strengths And Weaknesses:**

Strength:
1. Proposed method demonstrated strong empirical performance on a wide range of tasks, including challenging reasoning-based T2I on WISE benchmark.
2. It proposes a unified objective that jointly optimizes text-based reasoning and image generation, which is significant and challenging.
3.  The design of False-positive rectification is a novel contribution and addresses the problem of potentially misguiding refinement results.

Weakness.
1. Lack of comparison. While the concept of DVReward is novel, the author failed to compare against several latest VLM-based reward models such as HPSv3 and Unified Reward. In particular, Unified reward is also "decomposable" in that the VLM will also output individual scores for each object and check if they are present in the image. It is unclear from existing experiments if the improvements come from using a larger, more advanced VLM backbone Qwen3-VL-30B-A3B.
2. Author should evaluate the latency of their reward model compared with baselines (e.g. CLIP-based Pickscore). Although author claimed "the latency awaiting reward feedback can be reduced to a negligible level" by "asynchronously calling the reward model", this is not convincing without quantitive metrics and more detailed discussion of asynchronously scheduling.  (e.g. are models hosted on separate server? What is the capacity of those servers? If you have N training nodes, do you need N servers or only 1 server? What is the total overhead in terms of number of extra GPUs used to serve the reward model?)

---

> ### Author Rebuttal · Authors · 2026-03-31
>
> We thank the reviewer for recognizing the strong performance, unified objective design, and the novelty of the methods.
>
> &nbsp;
> ### **W1/Q2-1: Comparison with HPSv3 and UnifiedReward**
> ---
> The results show that HPSv3 and UnifiedReward could not achieve consistent improvements on downstream benchmarks which proves the effectiveness of DVReward compared to other VLM-based reward models.
>
> Besides, we want to emphasize that **the compared VIEScore uses the same MLLM as DVReward and observed heavy degradation** on GenEval (-2.3) which proves the non-trivial design of DVReward and the improvement of DVReward does not merely rely on more advanced MLLMs.
>
> |Reward Model|TIIF-S|TIIF-L|GenEval|
> |:------------|:------:|:------:|:-------:|
> |-(no RL)|75.2|78.6|84.0|
> |HPSv3|78.5|77.1|83.4|
> |UnifiedReward|**79.2**|77.3|83.7|
> |VIEScore|79.1|77.9|81.7|
> |**DVReward (ours)**|78.9|**79.5**|**85.0**|
>
> &nbsp;
> ### **W1-2: The effect of #questions in DVReward**
> ---
>
> We conduct experiments on DVReward with reduced question count. To preserve the completeness of
> evaluation coverage rather than randomly sampling from the original question set, we use Qwen3-235B-A22B
> to merge the full questions into maximum 10 core questions per sample.
>
> The results show inconsistent improvements: while comparable performance appears on TIIF-Bench, there is **a significant performance drop on GenEval, especially -11.3 on Spatial and -10.7 on Count.**
>
> We hypothesize that merging questions implicitly collapses multiple fine-grained verification points into one coarse question, resulting in a reward signal that cannot distinguish individual attributes (e.g., spatial relations, counting). **This degrades basic capabilities that benefit from atomic-level supervision, validating the importance of DVReward's fine-grained decomposition.**
>
> |#Questions|TIIF-S|TIIF-L|GenEval-Overall|GenEval-Spatial|GenEval-Count|
> |:-|:-:|:-:|:-:|:-:|:-:|
> |-|75.2|78.6|84.0|71.0|74.0|
> |10|78.2|**79.7**|81.1|63.0|71.8|
> |All|**78.9**|79.5|**85.0**|**74.3**|**82.5**|
>
> &nbsp;
> ### **W2: Latency of calling DVReward during training**
> ---
> We deploy one GPU per node to serve DVReward, resulting in a training-to-serving GPU ratio of 7:1.
> We compare the CLIP-based PickScore without deploy serving and our DVReward with deploy serving. We only record the bubble time of waiting for reward feedback.
> The results are calculated by running 100 steps training.
> With well-designed optimization, **DVReward achieves even lower bubble time than PickScore (9.72e-06s vs 4.19e-02s)**, despite requiring ~20 verification calls per sample.
> |Reward Model|Bubble Time(s)|
> |:-|:-:|
> |PickScore|4.19e-02±1.29e-03|
> |DVReward|**9.72e-06±3.63e-07**|
>
> &nbsp;
> ### **Q1: Details of "asynchronously scheduling" and optimization of reward serving.**
> ---
> The verification during RL training **doesn't block the training process** by applying high performance serving engine (SGLang) and the designed **"asynchronous scheduling"**.
> We clarify the reasoning why DVReward is fast to respond and how we reduce the latency into a minimum step-by-step:
> - The multiple calling of verification for each sample **share the same prefix of system prompt and image** that allow **reuse kv cache without recompute the image features and the prefilling of LLM**.
> - The additional cost per question verification is extremely minimal: **only prefilling ~20 tokens per question** ( a few hundred tokens total of all questions in one sample) and **only decoding one Yes/No token**. All questions can be processed in parallel on the reward server.
> - The verifier (Qwen3VL-30B-A3B) with only 3B active parameters enables high throughput (**over 4K tokens/s in prefilling, over 500+ running requests** on one A100), enabling very low verification latency.
> - **Decentralized reward servers eliminate cross-node communication bottleneck** in multi-node training. Centralizing the reward server on a single node (8 GPUs) would require all nodes to send hundreds of requests with images in a second over the network. Instead, we deploy one reward server (1 GPU) per node, keeping all requests local.
> - Our **asynchronous scheduling** eliminates idle time in the training process. During RL training, a batch is split into 4 mini-batches for sequential rollout. Each mini-batch is submitted to the reward server immediately after rollout. Instead of synchronously waiting after rollout, **rewards are collected only after all mini-batches complete and before the current mini-batch training.** This overlaps reward computation with subsequent rollouts and previous minibatch's training, eliminating the bubble time.
>
> We provide a quantitative bubble time (waiting the reward response) analysis.
> After carefully designing, the bubble time can be reduced to a negligible level and our **DVReward is practical in modern RL training.**
> |Centralized Serving|Asynchronous Scheduling|Bubble Time(s)|
> |:-:|:-:|:-:|
> |❌|❌|40.8±3.42|
> |✅|❌|7.28±0.73|
> |✅|✅|**9.72e-06±3.63e-07**|

---

> > ### Author Rebuttal · Reviewer_a3xb · 2026-04-03
> >
> > Concerns are fully resolved, especially regarding the latency. Hence I'm increasing the score

---

> > > ### Author Response · Authors · 2026-04-03
> > >
> > > We are glad to hear that all concerns have been addressed.
> > > We thank the reviewer for the valuable suggestions during the rebuttal.
> > > We will update the experiments and discussions in the revision.

---

### Official Review · Reviewer_8UXc · 2026-03-12

**Soundness:** 2
**Presentation:** 2
**Significance:** 2
**Originality:** 2
**Overall Recommendation:** 3
**Confidence:** 4

**Summary:**

The paper proposes AlphaGRPO, an RL-based training framework that extends GRPO to unified multimodal models that combine autoregressive (AR) reasoning text generation with diffusion-based image generation. A central component is DVReward (Decompositional Verifiable Reward), where an LLM decomposes a user request into verifiable sub-questions and an MLLM verifies these sub-questions on generated images to produce a scalar reward. The method is claimed to improve reasoning text-to-image generation and self-reflective refinement.

**Compliance With Llm Reviewing Policy:**

Affirmed.

**Key Questions For Authors:**

1. What exactly do the authors mean by “without external knowledge injection,” and how is DVReward’s reliance on LLM/MLLM not a form of external knowledge supervision?
2. Do the authors have a realistic cost/latency/throughput analysis, and a path to making DVReward practical without massive external models and heavy sampling (e.g., G=14)?
3. What evidence (OOD prompts, human evaluation, adversarial checks) shows the method generalizes beyond synthetic prompts and is robust to verifier gaming/reward hacking?
4. Other items mentioned above as well.

**Limitations:**

See mentioned above

**Strengths And Weaknesses:**

Main Concerns：
1. The abstract claims “without relying on external knowledge injection,” but it is unclear what this means in practice. If DVReward depends on an LLM/MLLM pipeline for decomposition and verification, this seems equivalent to using an external knowledge judge/teacher, rather than purely “unlocking intrinsic potential” of the base generative model.

2. DVReward appears difficult to deploy in real-world settings due to high inference cost and system complexity: per sample it requires LLM decomposition (often multiple questions), MLLM verification per question, and group sampling with G=14 trajectories per prompt. This seems impractical, especially when relying on very large models (e.g., 30B/235B) and their pre-existing capabilities.

3. Training prompts are synthesized via an LLM following the TIIF-Bench taxonomy (39 task types), which may bias the model toward template-like, synthetic prompt distributions and limit generalization to real user prompts.

4. Optimizing for “passing verifier QA” risks reward hacking: the model may learn to generate images that are easiest for the verifier to judge as correct, which may not reflect true semantic correctness or human preference.

5. Setting KL coefficients 𝛽𝐴𝑅 and βFlow to 0 is unusual and potentially risky, as it removes a common constraint preventing policy drift and training instability; the rationale and stabilization mechanism are unclear.

---

> ### Author Rebuttal · Authors · 2026-03-31
>
> We appreciate the detailed and thorough review. We address each concern below.
>
> &nbsp;
> ### **W1/Q1: Reclaim "Without external knowledge injection"**
> ---
> Our intended distinction is that we enhance the multimodal generation ability via RL training **without an additional SFT stage to distill external knowledge from stronger teacher models**.
>
> Recent work [1] demonstrates that RL cannot exceed the pretrained model's intrinsic capabilities.
> Based on this distinction,
> - What we avoid: Prior UMM works (e.g., UniRL, IRG) **synthesize reasoning SFT data by using proprietary models (e.g., GPT-4o)** that explicitly distill the external knowledge. The performance gains originate from the external teacher.
> **AlphaGRPO applies RL directly to the pretrained model without any cold-start SFT** on RT2I and SRR tasks.
> - What we do: DVReward serves as a selector, not a distillation teacher. Since **the model can only rollout samples within its own pretrained distribution**, the reward model only shifts probability mass toward higher-quality samples within the existing distribution.
>
> > [1] Yue et al., "Does RL Really Incentivize Reasoning Capacity in LLMs Beyond the Base Model?" NeurIPS 2025
>
> &nbsp;
> ### **W2/Q2-1: DVReward deployment cost**
> ---
> We argue that DVReward is practical for modern RL training with proper system optimization:
> - **Question decomposition is done offline and does not increase training time**
> - **Online verification is fully overlapped with the rollout pipeline through asynchronous scheduling** and SGLang-based prefix KV cache reuse with decentralized serving. The average reward-waiting time is **reduced from 40.8s to 9.72e-06s**. Please see our response to Reviewer a3xb W2 for detailed analysis.
>
> &nbsp;
> ### **W2-2: Group Size**
> ---
> We clarify that large group sizes are common in image generation RL training. DanceGRPO uses G=12, Flow-GRPO uses G=24 for SD3.5 and G=16 for Qwen-Image. **AlphaGRPO achieves stable training with G=14, which is comparable or smaller than prior works.**
> Reducing group size while maintaining training stability remains an open challenge, which we leave for our future work.
>
> &nbsp;
> ### **W3/Q3: Template bias from TIIF-Bench taxonomy**
> ---
> We clarify that we only follow the TIIF-Bench task taxonomy to define compositional categories and **do not reuse any prompt templates or prompts from TIIF-Bench**. All training prompts are independently synthesized.
>
> More importantly, **our evaluation benchmarks already constitute OOD testing**. The training prompts, reward model and evaluation protocols are **entirely disjoint from the downstream benchmarks**. The consistent improvements across all of them already demonstrate strong OOD generalization.
>
> We further provide human evaluation on real-user prompts (see our response to Reviewer a6Kf W3/Q2). The results show that AlphaGRPO achieves a 10% higher win rate than the baseline, confirming robust generalization to real-world scenarios.
>
> &nbsp;
> ### **W4: Reward hacking risk**
> ---
> Reward hacking, where the model exploits the proxy reward while true quality degrades, is a well-recognized open challenge in RL, with no general solution to date [2, 3]. Any reward model is subject to this risk. As in our Introduction, the **key motivation** of DVReward is precisely this **designing a reward mechanism that applies a MLLM as reward model and generalizes beyond the training distribution**. It is inherently an anti-reward-hacking objective.
>
> Our experiments validate that DVReward achieves this: AlphaGRPO trained with **DVReward consistently improves across 6 OOD downstream benchmarks with different evaluation protocols**, i.e., GenEval (+1.0, detector+CLIP with rules), TIIF-Bench (+3.7/+0.9, GPT-4.1), DPG-Bench (+1.0, mPLUG-large) and GEdit (+0.52, GPT-4.1). In contrast, VIEScore (**holistic scalar scoring using the same MLLM as DVReward) degrades performance** on GenEval and TIIF-L (Table 5), demonstrating that not all MLLM-based rewards generalize to OOD downstream. DVReward's decomposition-then-verify design is the key differentiator. See our response to Reviewer a6Kf W3/Q2 for human evaluation on real user prompts.
> > [2] Black et al., "Training Diffusion Models with Reinforcement Learning" ICLR 2024
>
> > [3] Gao et al., "Scaling Laws for Reward Model Overoptimization" ICML 2023
>
> &nbsp;
> ### **W5: KL coefficients**
> ---
> Setting β=0 is well-precedented in recent works. DDPO (ICLR 2024 Oral) uses no KL regularization for diffusion model RL. DAPO (NeurIPS 2025) removes KL from GRPO entirely and Open-Reasoner-Zero (NeurIPS 2025) ablates KL and finds that KL mechanisms slow down training and consume additional resources. **Whether to use KL is an empirical design choice**, not a theoretical necessity.
>
> In our experiments, adding a KL penalty (β = 0.001) does not provide consistent gains and degrades performance on TIIF-L and GenEval.
> |β_AR|β_Flow|TIIF-S|TIIF-L|GenEval|
> |:-|:-:|:-:|:-:|:-:|
> |0|0|78.9|**79.5**|**85.0**|
> |0.001|0.001|**79.2**|78.7|84.1|

---

> > ### Author Rebuttal · Reviewer_8UXc · 2026-04-04
> >
> > Thank you for the rebuttal.
> > The explanation of “without external knowledge injection” remains unconvincing, because the method still relies on external LLM/MLLM components for decomposition and verification, which is difficult to distinguish from external knowledge supervision.
> > The response on practicality also does not fully resolve my concern, since reducing reward waiting time through asynchronous overlap does not address the underlying computational cost and system complexity of the full pipeline.
> > For these reasons, I maintain my score.

---

> > > ### Author Response · Authors · 2026-04-07
> > >
> > > Thank you for the acknowledgement. We address the two remaining concerns:
> > >
> > > &nbsp;
> > >
> > > ### **1. "Without external knowledge injection"**
> > > ---
> > >
> > > We think the reviewer might misunderstand our meaning in our rebuttal.
> > > We agree the phrasing is debatable and will update this to **"without cold-start SFT stage"**, which is more precise.
> > > We apologize for the confusion.
> > >
> > > However, **revising this phrasing does not affect the paper's core contributions and findings**:
> > > 1. AlphaGRPO targets multimodal generation GRPO training on native AR-Diffusion UMMs, covering reasoning T2I and self-reflective refinement, exhibiting high potential to extend to interleaved multimodal generation.
> > > 2. DVReward applies a general MLLM as a zero-shot reward model that **consistently improves across all OOD benchmarks**, while existing VLM-based reward models (VIEScore, HPSv3, UnifiedReward) fail to generalize consistently.
> > > 3. SRR training generalizes to T2I benchmarks without targeting them, achieving performance gains comparable to RT2I training and even better on image editing.
> > > 4. Training at 512 resolution consistently improves 1024 resolution performance.
> > >
> > > &nbsp;
> > > ### **2. Lower computational cost in DVReward and adaptable optimization for existing VLM-based human preference models**
> > > ---
> > >
> > > We respectfully argue that the reviewer may overestimate DVReward's cost relative to existing VLM-based human preference models.
> > > The computational cost of DVReward is **lower than current VLM-based reward models** for the following reasons:
> > >
> > > (a) **No human annotation** Existing VLM-based reward models also need a pretrained MLLM as the backbone, and **require large-scale human preference annotations** for further training, e.g., **UnifiedReward uses 798K annotated data** and **HPSv3 collects 1.17M pairwise human annotations**.
> > >
> > > (b) **No training cost.** Before they can provide stable reward signals, current VLM-based reward models **require multi-stage training including SFT and RL, all incurring higher computational cost and complex training design overhead**. Instead, **DVReward is an out-of-the-box method** that leverages current open-source MLLMs as zero-shot generalizable reward models, which is a non-trivial contribution (VIEScore, using the same MLLM, fails to achieve OOD generalization).
> > >
> > > (c) **Light decode cost.** Recent VLM-based reward models, e.g., UnifiedReward-series, increasingly adopt a "think-then-verify" paradigm, requiring the model to generate long reasoning chains (hundreds of tokens) before producing a score. **The heavy decoding in current VLM-based reward models brings significant latency and slows RL training.** In contrast, DVReward can perform the decomposition offline (241 GPU hours for 20K prompts, only 7% of the AlphaGRPO RL training cost) and only decodes **a single Yes/No token per question**. Moreover, our optimization for DVReward serving is also applicable to those VLM-based reward models (e.g., UnifiedReward).
> > >
> > > <!-- 241 hours for 2w prompts. 3072 hours in alphagrpo training. -->
> > >
> > > | | Cost | UnifiedReward-Series | HPSv3 | **DVReward (ours)** |
> > > |:--|:--:|:--:|:--:|:--:|
> > > | Human annotation | Extremely High | >798K | 1.17M | **No** |
> > > | Additional training | High | SFT + RL | SFT | **No** |
> > > | Decomposition | Low | N/A | N/A | **Offline (one-time)** |
> > > | Decoding in online RL training | Medium | Hundreds of tokens | Single score | **1 token × ~20 questions in parallel** |
> > >
> > > Besides, compared to current VLM-based reward models, e.g., HPSv3 and UnifiedReward, our DVReward achieves consistent improvements on downstream benchmarks (please refer to Reviewer a3xb W1/Q2-1).
> > >
> > > In summary, DVReward only requires one-time offline decomposition and a lightweight online verification with **zero human annotation cost, zero training cost, and strong OOD generalization**, making it more practical than existing alternatives. **We sincerely hope the reviewer could recognize our contributions and reconsider the scoring.**

---

### Official Review · Reviewer_a6Kf · 2026-03-12

**Soundness:** 3
**Presentation:** 3
**Significance:** 3
**Originality:** 2
**Overall Recommendation:** 4
**Confidence:** 4

**Summary:**

This paper proposes AlphaGRPO, a framework for improving unified multimodal models that combine autoregressive reasoning and diffusion-based image generation. The authors apply GRPO to a unified trajectory consisting of reasoning tokens followed by diffusion generations. In addition, the paper introduces DVReward that decomposes prompts into semantic and quality questions and uses a multimodal LLM verifier to compute reward signals. Experiments are conducted on several text-to-image benchmarks and an image editing benchmark , where authors report improved performance over the backbone model BAGEL.

**Compliance With Llm Reviewing Policy:**

Affirmed.

**Final Justification:**

During the rebuttal, the authors presented extended experiments on baseline comparisions and sensivity analysis to different rewards, which helps address my main concerns. While I retain my reservations about the originality and novelty of the proposed method, the consistent empirical improvements and robustness of the framework indicate a meaningful strength on this task.

Hence, I raised the soundness score from 2 to 3, and the overall recommendation from 3 to 4 to reflect my assessment after the discussion.

**Key Questions For Authors:**

- can you compare to some of the existing RL training pipelines for diffusion models, for example, PPO for Diffusion [1] and/or Flow-based RL baselines such as Flow-GRPO[2]?
- It would be helpful to evaluate whether improvements are robust under different reward models or evaluation protocols, e.g. alternative reward model, evaluation metrics independent of the reward model and human evaluations.

[1] Black, Kevin, Michael Janner, Yilun Du, Ilya Kostrikov, and Sergey Levine. "Training diffusion models with reinforcement learning." arXiv preprint arXiv:2305.13301 (2023).
[2] Liu, Jie, Gongye Liu, Jiajun Liang, Yangguang Li, Jiaheng Liu, Xintao Wang, Pengfei Wan, Di Zhang, and Wanli Ouyang. "Flow-grpo: Training flow matching models via online rl." arXiv preprint arXiv:2505.05470 (2025).

**Limitations:**

yes

**Strengths And Weaknesses:**

Strengths:
- this paper extends RL methods to unified multimodal generation, which is a worthwhile research direction.
- the evaluation is done on several commonly used benchmarks for text-to-image alignment and editing
- the proposed DVReward is practical for obtaining more stable reward signals.


Weaknesses:
- my most significant concern of the paper is the lack of strong RL baselines. The main comparisons are performed against the backbone model BAGEL, which does not involve reinforcement learning. there exists several prior work on RL optimization for diffusion models and text-to-image generation. it is necessary to compare with some of these baselines to determine whether the observed improvements are due to the specific design of AlphaGRPO or simply due to applying RL training.
- the technical novelty is rather limited, as the paper primarily integrates several existing components: GRPO, diffusion trajectory optimization, reward models derived from multimodal LLMs. While the combination is interesting, each component has been explored in prior work, which makes the conceptual novelty incremental.
- the reward relies on a powerful multimodal LLM to verify semantic questions about generated images. this could be an issue in particularly when the evaluation benchmarks also measure semantic alignment between prompts and images, leading to the optimization being overfitting to the preferences of the reward model rather than generation quality.

---

> ### Author Rebuttal · Authors · 2026-03-31
>
> We thank the reviewer for recognizing that extending RL to unified multimodal generation is a worthwhile direction and that DVReward is practical.
>
> &nbsp;
> ### **W1/Q1: Lack of strong RL baselines**
> ---
>
> We conduct experiments with Flow-GRPO on BAGEL following the official implementation.
> The results reveal that Flow-GRPO fails to generalize to downstream benchmarks with degradation on TIIF-Bench Long prompt and Geneval.
>
> | Method              | TIIF-S | TIIF-L | GenEval |
> |:--------------------|:------:|:------:|:-------:|
> | BAGEL (no RL)       | 75.2   | 78.6   | 84.0    |
> | Flow-GRPO on BAGEL  | 75.7   | 77.0   | 83.7    |
> | AlphaGRPO (RT2I)    | 78.9   | **79.5**   | **85.0**   |
> | AlphaGRPO (SRR)     | **79.1**   | **79.5**   | 84.2    |
>
> We note a key structural difference: Both DDPO and Flow-GRPO optimize only the diffusion flow, while AlphaGRPO **jointly optimizes reasoning discrete text + continuous visual flow** in a unified trajectory. This joint optimization is what enables Reasoning Text-to-Image Generation and Self-Reflective Refinement that DDPO and Flow-GRPO fundamentally cannot support, as they lack text reasoning optimization.
>
> &nbsp;
> ### **W2: Technical novelty is limited**
> ---
>
> We respectfully disagree that the novelty is limited. Our work makes several non-trivial contributions and findings:
>
> 1. We proposed AlphaGRPO that targets multimodal generation GRPO training on native AR-Diffusion UMMs. The two tasks we study, reasoning T2I and self-reflective refinement, are core **sub-tasks of interleaved multimodal generation**, demonstrating AlphaGRPO's strong potential to extend to full interleaved generation.
> 2. We propose DVReward, applying a general MLLM as a zero-shot reward model via two stages decomposition and verification. We find that the holistic scoring method (e.g., VIEScore) that leverage MLLMs to give scalar scores would degrade performance on OOD downstream benchmarks, while **DVReward consistently improves the model across all benchmarks.**
> 3. We discover that despite not training on T2I tasks, our AlphaGRPO (self-reflective refinement) achieves comparable or even superior improvements on downstream T2I benchmarks compared to directly training on reasoning T2I.
> 4. We find that AlphaGRPO **training at 512 resolution consistently improves on 1024 resolution**, demonstrating that the learned alignment is resolution-agnostic and further alleviates the resource consumption requirement for high-resolution RL training.
>
> &nbsp;
> ### **W3/Q2: Reward overfitting to MLLM preferences**
> ---
>
> We provide three lines of evidence against reward overfitting:
>
> - **OOD downstream benchmarks generalization**: The training prompt set is different from the downstream benchmarks, AlphaGRPO improves on image generation benchmarks and image editing benchmark. All these benchmarks use different evaluation protocols or different target tasks:
>     - Generalize to OOD image generation benchmarks: GenEval (+1.0, detector + CLIP with rules), TIIF-Bench (+3.7/+0.9, GPT-4.1), DPG-Bench (+1.0, mPLUG-large)
>     - **Generalize to OOD editing benchmark without training on editing task**: GEdit (+0.52, GPT-4.1).
> - **Human evaluation**: We conduct a human evaluation on 200 real-user prompts from the Pick-a-Pic dataset. Three annotators evaluate each pair on prompt following, perceptual quality, and overall preference. The results show that AlphaGRPO is consistently preferred over BAGEL across all dimensions:
> | Dimension | BAGEL Win | Tie | AlphaGRPO Win |
> |:----------|:---------:|:---:|:------------:|
> | Overall | 30.5% | 29.0% | **40.5%** |
> | Prompt Following | 25.5% | 38.0% | **36.5%** |
> | Perceptual Quality | 39.0% | 18.0% | **43.0%** |
>
> - **Consistent improvement on Additional Reward Models**: We additionally enlarge the real user prompts set to 1K and use PickScore (scale to [0,1]), ClipScore and AestheticScore to evaluate the performance of AlphaGRPO. The results show that AlphaGRPO achieves **consistent improvements** on all reward models.
> | Reward Model | BAGEL | AlphaGRPO |
> |:----------|:---------:|:---------:|
> | AestheticScore | 5.9583 | **5.9624** |
> | ClipScore | 0.8984 | **0.9102** |
> | PickScore | 0.8208 | **0.8246** |

---

> > ### Author Rebuttal · Reviewer_a6Kf · 2026-04-04
> >
> > I thank the authors for the added experiment results.  I have a follow up question:
> >
> > - Regarding W1, can the authors provide on some intuitions why RL baselines fails on the two benchmarks? related to this, the parameter $\lambda$ in eqn.4 aims to balance the joint optimization of text and visual flow, but no detailed analysis/study was conducted on that. It could help understand how much gains this joint training objective helps over visual flow only RL.

---

> > > ### Author Response · Authors · 2026-04-07
> > >
> > > Thank you for the follow-up questions.
> > >
> > > &nbsp;
> > >
> > > ### **1. Why does visual-only RL fall behind AlphaGRPO**
> > > ---
> > >
> > > We think the possible reason is that if we only apply visual flow RL, the model would easily overfit to hidden format patterns in the training prompts, making it difficult to generalize to OOD downstream benchmarks with different prompt patterns. Since AlphaGRPO targets reasoning text-to-image generation and self-reflective refinement, both of which require generating thinking text before image generation, **this provides diverse text conditions during RL training, preventing the model from overfitting to hidden patterns and improving prompt following performance on OOD benchmarks**.
> > >
> > > &nbsp;
> > > ### **2. Analysis of λ**
> > >
> > > We initially set λ=0.2 because in our early experiments on the SRR task, higher text weights caused training instability and collapse. For consistency, we applied the same λ=0.2 across both RT2I and SRR tasks. Following the reviewer's suggestion, we revisited this choice on the RT2I task and conducted an experiment with λ=1.0, which achieves better performance. This demonstrates that **increasing the text optimization weight helps the model generalize to downstream T2I benchmarks**.
> > >
> > >
> > > | λ | TIIF-S | TIIF-L | GenEval |
> > > |:--|:--:|:--:|:--:|
> > > | 0.2 (default) | 78.9 | 79.5 | 85.0 |
> > > | 1.0 | **79.0** | **79.6** | **85.7** |
> > >
> > > Due to the time and resources limitation, detailed ablation of λ on more values and all tasks would be updated in the revision.
> > >
> > > We hope the above analysis and new experiments could address the reviewer's concerns. We sincerely appreciate the constructive feedback which has helped improve our work.

---

### Decision · Program_Chairs · 2026-04-30

**Decision:**

Accept (regular)

**Comment:**

In this paper, the authors proposed AlphaGRPO, an RL-based framework that extends GRPO to UMM  (autoregressive reasoning text generation with diffusion-based image generation), with one of the key contributions as Decompositional Verifiable Reward, where user's prompt is decomposed into verifiable sub-questions by LLM, each of which is verified by an MLLM.

After rebuttal, most reviewers' concern have been addressed and provided positive scores. Only reviewer 8UXc has the score "weak reject" with two major concerns: 1) Using external LLM/MLLM components is also "external knowledge injection" 2) DVReward would complicate the system. Both points are valid, however, not a deal breaker.  For 1),  the authors should indeed modify some claims in the paper, but its main contributions are not affected. For 2), the advantages that DVReward does not need human annotated data and can be computed parallelly may bring more benefits.

So I will recommend " Weak Accept".